# Adaptive mechanisms facilitate robust performance in noise and in reverberation in an auditory categorization model

Satyabrata Parida [1,2], Shi Tong Liu[3] & Srivatsun Sadagopan [1,2,3,4 ✉]

For robust vocalization perception, the auditory system must generalize over variability in vocalization production as well as variability arising from the listening environment (e.g., noise and reverberation). We previously demonstrated using guinea pig and marmoset vocalizations that a hierarchical model generalized over production variability by detecting sparse intermediate-complexity features that are maximally informative about vocalization category from a dense spectrotemporal input representation. Here, we explore three biologically feasible model extensions to generalize over environmental variability: (1) training in degraded conditions, (2) adaptation to sound statistics in the spectrotemporal stage and (3) sensitivity adjustment at the feature detection stage. All mechanisms improved vocalization categorization performance, but improvement trends varied across degradation type and vocalization type. One or more adaptive mechanisms were required for model performance to approach the behavioral performance of guinea pigs on a vocalization categorization task. These results highlight the contributions of adaptive mechanisms at multiple auditory processing stages to achieve robust auditory categorization.

[1] Department of Neurobiology, University of Pittsburgh, Pittsburgh, PA, USA. [2] Center for the Neural Basis of Cognition, University of Pittsburgh, Pittsburgh, PA, USA. [3] Department of Bioengineering, University of Pittsburgh, Pittsburgh, PA, USA. [4] Department of Communication Science and Disorders, University of Pittsburgh, Pittsburgh, PA, USA. ✉email: vatsun@pitt.edu

To maintain robust auditory perception, especially of communication sounds such as vocalizations (calls), animals and humans must generalize over the tremendous variability in the production of these sounds and the variability imposed by the listening environment. Production variability includes both trial-to-trial and subject-to-subject variability that is inherent in sound production. Environmental variability refers to a variety of acoustic degradations such as the addition of background noise and reverberation. Call categorization thus requires a many-to-one mapping operation, where diverse acoustic inputs are mapped onto a small number of behaviorally relevant categories. How this categorization operation is implemented by the neural circuitry of the auditory system is a central question in auditory neuroscience. Existing auditory encoding models are not well-suited to address this question. For example, many auditory pathway models adopt an engineering approach, and process inputs using experimenter-defined spectral, temporal, or modulation filter banks, which are biologically inspired;[1,2] however, these models focus on stimulus encoding and have not been tested in categorization tasks. Another class of models, which are based on deep neural networks, can achieve robust auditory categorization performance that can approach human performance levels[3,4]. However, these complex networks offer limited biological interpretability, and an intuitive understanding of what stimulus features are used to generalize over a given category, and how these stimulus features are biologically computed, is elusive. Additionally, the latter class of models requires a large amount of training data (typically of the order of millions of data points). To strike a balance between biological interpretability and categorization performance, we have previously proposed a hierarchical model that learns informative, intermediate-complexity features that can generalize over within-category variability and accomplish categorization[5]. This model can achieve robust production invariance with a limited training set (of the order of hundreds of data points). However, in its simplest implementation, model performance degrades when environmental variability is introduced to model inputs. In the present study, we characterize model performance for a few common types of environmental degradations and extend the model to include several adaptive neural mechanisms that may aid auditory categorization in such conditions.

The hierarchical model consists of three stages (Fig. 1). The first stage is a dense spectrotemporal representational stage, which is the output of a phenomenological model of the cochlear filter bank[6]. The second stage consists of a set of sparsely active feature detectors (FDs). Each FD has a spectrotemporal receptive field (STRF), which corresponds to the stimulus feature that the FD is tuned to detect, and a static output nonlinearity (threshold). Optimal feature tuning for performing specific call categorization tasks, and optimal thresholds to maximize classification performance, are learned using greedy search optimization and information-theoretic principles[5]. Optimal FDs typically exhibit bandwidths of around 2 octaves and durations of around 200 ms, and are thus tuned to detect spectral structures and temporal sequences of this scale in the stimuli. The final 'voting' stage of the model obtains the evidence for the presence of a call category by combining the outputs of each category's FDs, weighted appropriately. Note that, to output the final call category in a go/no-go paradigm, the voting stage can be implemented as a winner-take-all algorithm[7]. The temporal sequencing of the detected features, which may be important for selectivity for longer time scale (> 200 ms) stimulus variations, is presently not considered in the model. We have previously shown that this hierarchical framework can achieve high accuracy in categorizing calls across multiple species, and that model performance in categorizing natural and manipulated guinea pig (GP) calls

mirrors GP behavior[7]. In addition, we showed that in guinea pigs (GPs), subcortical and layer 4 neurons in the primary auditory cortex (A1) show simple frequency-tuned receptive fields and dense activities consistent with the spectrotemporal representation stage of the hierarchical model, whereas layer 2/3 neurons in A1 show complex feature-tuned receptive fields and sparse activities consistent with the FD stage[8]. This observation is consistent with other studies that suggest that receptive field complexity increases between A1 layer 4 and layer 2/3[8–10], and that sound category information may be encoded by neurons in the superficial layers of A1[11,12].

But the model is explicitly trained to generalize over production variability in calls, which can be conceptualized as spectrotemporal variations (limited by biological constraints) around an archetypal call for each category. This variability occurs on a trial-to-trial time scale. In contrast, environmental variability is not a function of call type, is not limited by biological capabilities, and typically varies at longer time scales. Thus, the fundamental strategy employed by the model to generalize over production variability, that of detecting informative features, is unlikely to also confer resilience to environmental variability. Therefore, in this study, we characterized model performance in two challenging conditions: with additive white Gaussian noise and in reverberant settings. We found that the basic model failed to generalize over these forms of environmental variability. Therefore, we extended the model to include several neural mechanisms that are known or have been hypothesized to improve neural coding in degraded environments. First, condition-specific training can improve the performance of computational models[13,14] as well as humans and animals[15]. We implemented condition-specific training by augmenting the training data set with degraded calls (i.e., calls that were affected by noise and/or reverberation). Second, contrast gain control, which refers to adaptive changes to neural tuning and activity levels to match incoming sound statistics, has been demonstrated in several stations along the ascending auditory pathway[16–22]. We implemented a version of contrast gain control by adapting neural responses in the dense spectrotemporal stage to the mean and standard deviation of population activity in that stage. Finally, recent studies have shown that attention-mediated feedback can control the excitability of cortical neurons to aid performance in challenging listening environments[23,24]. Specifically, arousal-related noradrenergic input from locus coeruleus[25,26] as well as cholinergic inputs from the basal forebrain[27–29] can increase the excitability of cortical principal neurons, typically by disinhibiting them by suppressing the activity of cortical inhibitory interneuron subtypes[30]. We modeled one possible implementation of this top-down pathway, by altering FD thresholds to modulate FD excitability. While all three mechanisms generally improved model performance both in noise and in reverberation, the trends of benefit varied across degradation type (noise or reverberation) as well as call type, which suggests that the auditory system may differentially rely on these mechanisms based on the degradation type and the spectrotemporal properties of the sound. These results demonstrate that multiple adaptive mechanisms acting at multiple auditory processing stages are necessary to achieve robust auditory categorization.

## Results

The computational mechanisms described in this manuscript extend a previously published model of auditory categorization[5]. For ease of reading, we begin by briefly summarizing the core details of this model (also see Methods). Model feature detectors (FDs) were trained to optimally categorize one conspecific call type (within-class) from all other call types (outside-class). To do

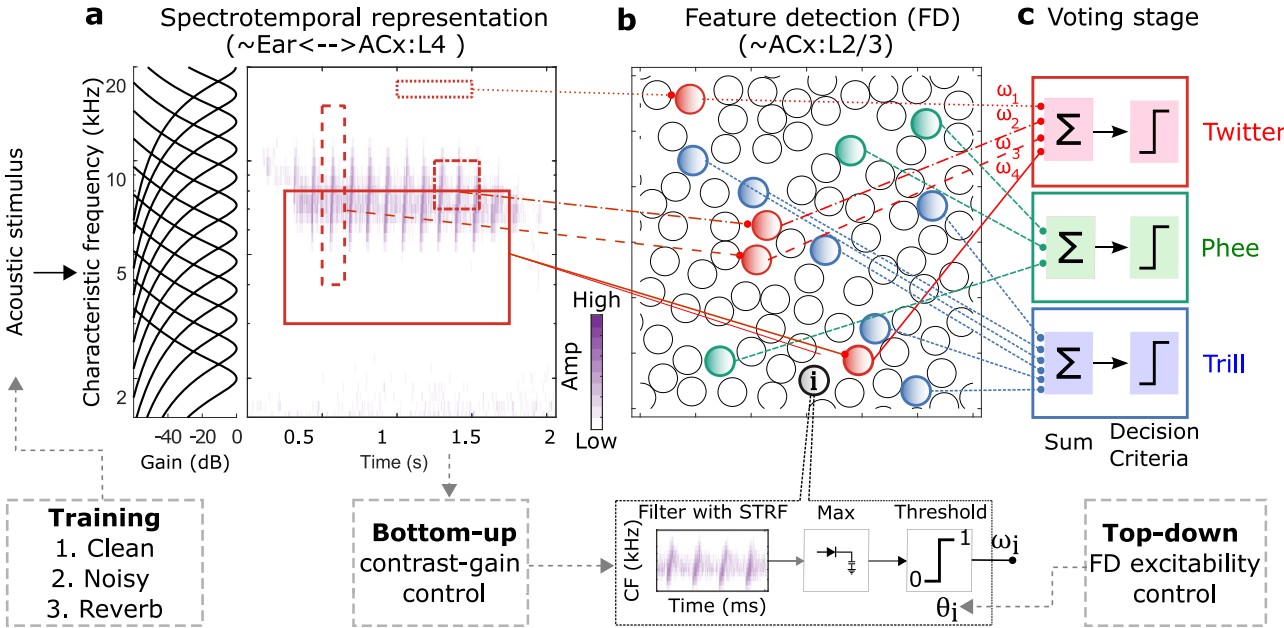

**Fig. 1 Hierarchical structure of the computational model.** The core model consisted of a dense spectrotemporal stage, a sparse feature detection stage, and a voting stage. **a** An acoustic stimulus was filtered by a cochlear filter bank to obtain a dense spectrotemporal representation of the stimulus, called a cochleagram. **b** The second stage had call-specific feature detectors (FDs) modeled as a spectrotemporal receptive field followed by a threshold. Template matching by cross-correlation (limited to the bandwidth of the FD) was used to obtain an FD $V_m$ response, and the $V_m$ response was compared with the threshold to obtain a binary output. Each call type had a set of most informative FDs, whose STRFs and thresholds were learned during model training. **c** Finally, in the voting stage, the weighted outputs of the FDs for a given call type were combined to form the final response of the model. In this study, in addition to these stages, we extended the model to include three mechanisms (boxes with dashed lines): (1) condition-specific training, (2) contrast gain control, and (3) top–down modulation of the excitability of FDs. ACx auditory cortex, Amp amplitude, CF Characteristic frequency, FD feature detector, STRF spectrotemporal receptive field.

so, a large number of random features, which served as candidate STRFs that the FDs might detect, were first generated by randomly selecting small portions of within-class cochleagrams with random center frequency, bandwidth, onset time, and duration. For each candidate FD, an incoming cochleagram was convolved with its STRF (restricted to the bandwidth of the STRF) to obtain the membrane potential response (or $V_m$ response). The maximum of the $V_m$ response was compared with the FD's threshold (learned as described next) to obtain the FD binary response or FD output, where FD output = 1 if the maximum of $V_m$ response ≥ threshold. To learn the FD threshold, we first constructed distributions of its $V_m$ response maximum for within-class calls as well as outside-class calls and set the FD threshold to the value that maximized classification merit, as quantified by the mutual information between the stimulus class and the FD output. The FD weight was set to the log-likelihood of classification. From this initial set of ~5000 candidate FDs, we employed an iterative greedy search algorithm[31] to obtain the most informative set of features (MIF set) for the categorization of each call type. To do so, we sequentially added candidate FDs to the MIF set as long as candidate MIFs were not similar (in an information-theoretic sense) to existing members of the MIF set and the classification performance of the MIF set continued to improve. In the final voting stage, outputs of FDs in the MIF set were weighted (by the learned weight) and summed to obtain the final output of the categorization model. Five separate instantiations of models (with non-overlapping MIF sets) were trained for each call type to assess training convergence and performance reliability. Models were tested using a new set of within-class and outside-class call types that were not used in training and model performance was quantified using the sensitivity index d prime (d′).

Model performance was quantified for categorizing calls from two different species: marmosets and guinea pigs (Fig. 2). Target marmoset calls consisted of the twitter, trill, and phee calls. All three marmoset calls shared similar long-term spectra but varied in their short-term spectrotemporal properties[32] (e.g., frequency versus amplitude modulation). Target guinea pig calls consisted of the chut, wheek, rumble, and whine calls. The guinea pig calls varied in both their long-term spectra (e.g., the wheek had higher frequency content than the others) as well as in other spectro-temporal properties[33]. While vocalization recordings from each species typically had high (>15 dB) SNR, recordings contained some colony noise, which had low-pass spectra. As colony noise affected all call types of either species similarly, it is unlikely to systematically affect the results presented here (other than perhaps leading to slightly poorer categorization performance). Therefore, we did not use any filters to denoise the recordings. The performance of various models was tested in several noisy and reverberant (e.g., Fig. 2) conditions. For noisy conditions, we added white Gaussian noise down to signal-to-noise ratios (SNRs) of −18 dB. For reverberant conditions, we convolved calls with impulse responses obtained from four environmental recordings and four simulated room reverberations (Supplementary Figure S1, also see Methods). It is important to note that the effect of background noise is additive, whereas the effect of reverberation is convolutional. Therefore, these manipulations affect call types differently. For example, reverberation minimally affects tonal calls (e.g., the guinea pig wheek or marmoset phee), but is severely detrimental to calls with fast frequency or temporal modulations (e.g., the guinea pig whine or the marmoset trill) by smearing spectrotemporal cues at short time scales.

**Condition-specific training improved model performance, but these benefits were typically limited to the same condition.** Previous computational and behavioral studies have shown that

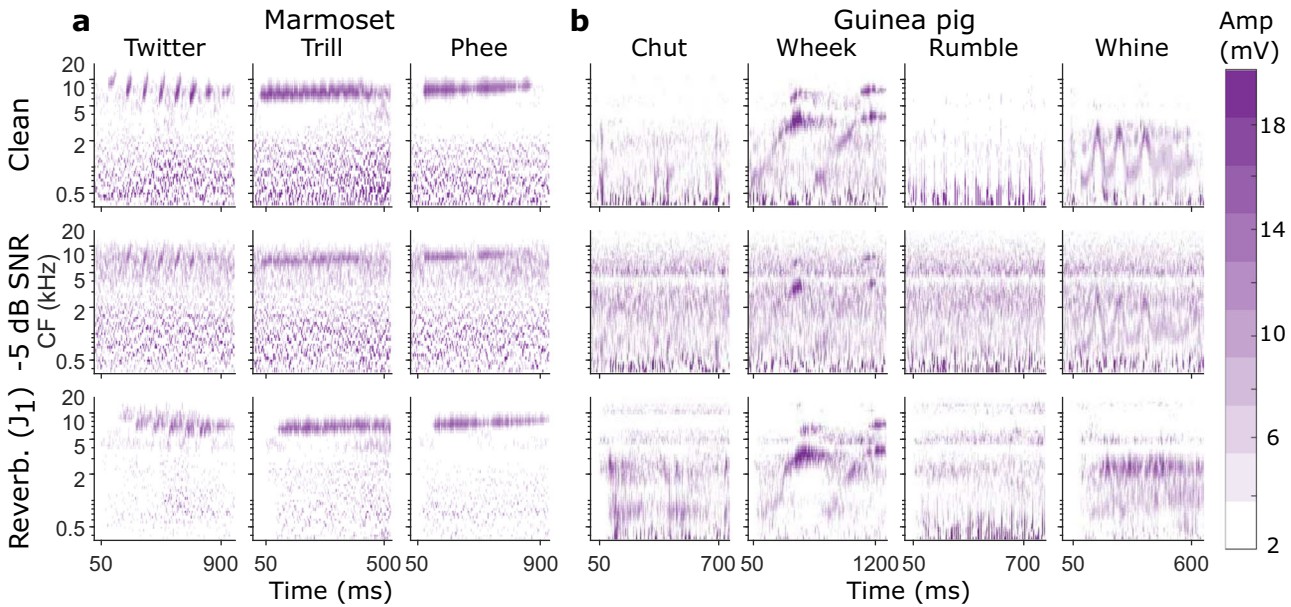

**Fig. 2 Effects of noise and reverberation on the spectrotemporal representation of exemplar marmoset and guinea pig calls. a** Cochleagrams of exemplar calls of the three major marmoset call types in quiet, at −5 dB SNR using white Gaussian noise, and in a reverberant condition. The three major marmoset call types are the twitter (characterized by repetitive rising frequency sweeps), the trill (predominantly frequency modulated), and the phee (a tonal call). All three calls have similar long-term spectra. **b** Same as **a** but for four major guinea pig call types, which are the chut (short-duration rising low-frequency sweep), the wheek (long-duration rising high-frequency sweep), the rumble (predominantly amplitude modulated), and the whine (predominantly frequency modulated). Both marmoset and guinea pig calls can contain multiple harmonics of the fundamental frequency trajectory. Color map corresponds to cochleagram amplitude in mV. Amp amplitude, CF center frequency, SNR signal-to-noise ratio.

categorization performance is better when training and testing conditions are the same (e.g., trained and tested using noisy calls) compared to when testing is done in a different condition (e.g., trained with clean calls and tested with noisy calls)[13–15]. To determine whether these results also applied to our hierarchical model, we trained and tested the model on calls degraded by noise or reverberation and compared its performance to the performance of the model trained only on clean calls. Condition-specific training significantly improved model performance both in noise and in reverberation for both species (Fig. 3, Supplementary Table S1). Note that Fig. 3 (and the following figures) only show two call types per species for brevity; however, statistics (in all tables) are reported based on all call types (three for marmosets, four for guinea pigs). Compared to models trained on clean calls and tested with noisy calls (black lines in Fig. 3a, b), training models with noisy calls (red lines) significantly improved performance when models were tested with noisy calls (Fig. 3a, b). Interestingly, the models trained on noisy calls sometimes performed slightly worse to categorize clean calls compared to the models that were trained on clean calls (e.g., Twitter and Chut, see data point marked '∞' in Fig. 3a, b).

To test whether these benefits translated across conditions, we tested the reverberation-trained model on noisy calls (green lines in Fig. 3a, b) and vice versa (red lines in Fig. 3c, d). Compared to same-condition training, across-condition training led to substantially lower improvement (indicated by lower $\chi^2$ values in Supplementary Table S1) over the clean-trained model for call types of both species (i.e., marmosets and guinea pigs). In fact, across-condition training sometimes resulted in poorer performance than the original model [e.g., noise-trained Wheek model when tested in reverberation, $\chi^2(1) = 33.9, p = 5.9 \times 10^{-9}$]. Overall, these results show that condition-specific training can improve auditory categorization performance, but these benefits do not generalize well to other conditions.

As a final model to assess the effect of training, we trained a model on a mix of noisy and reverberant conditions and hypothesized that its performance in noisy and reverberant test conditions would lie between the performances of models with same-condition and across-condition training. Surprisingly, the model trained on mixed conditions generally performed better than models with same-condition training (Supplementary Table S1, higher $\chi^2$ for mixed training compared to same-condition training). Overall, these results suggest that a model trained on multiple conditions can outperform models trained on specific conditions.

Next, we tested whether FD properties were systematically different for models trained using noisy calls or reverberant calls compared to the model trained using clean calls. We considered the duration, center frequency, bandwidth, reduced kurtosis, and threshold of FDs. We found little systematic difference for both marmoset call types (Supplementary Figure S2, Supplementary Table S2) and guinea pig call types (Supplementary Figure S3, Supplementary Table S3), as indicated by $\eta_P^2$ values, which were less than or equal to 0.05. The exceptions were feature duration, which was significantly longer (only for marmoset calls), and threshold, which was significantly lower, for FDs in noise-trained models. Intuitively, an increase in feature duration is expected because integration over longer durations might be necessary to overcome the degradation of features by noise. But a reduction (large and significant for marmoset calls, small and significant for guinea pig calls) in FD threshold was not expected because noise is traditionally assumed to increase the mean and reduce the standard deviation of neural activity[18] (but see potential explanation in the next section). Overall, the spectrotemporal properties of the FDs were largely unaffected by training in noise or reverberation.

Training in different listening conditions is biologically realistic. In both humans and animals, the acquisition of

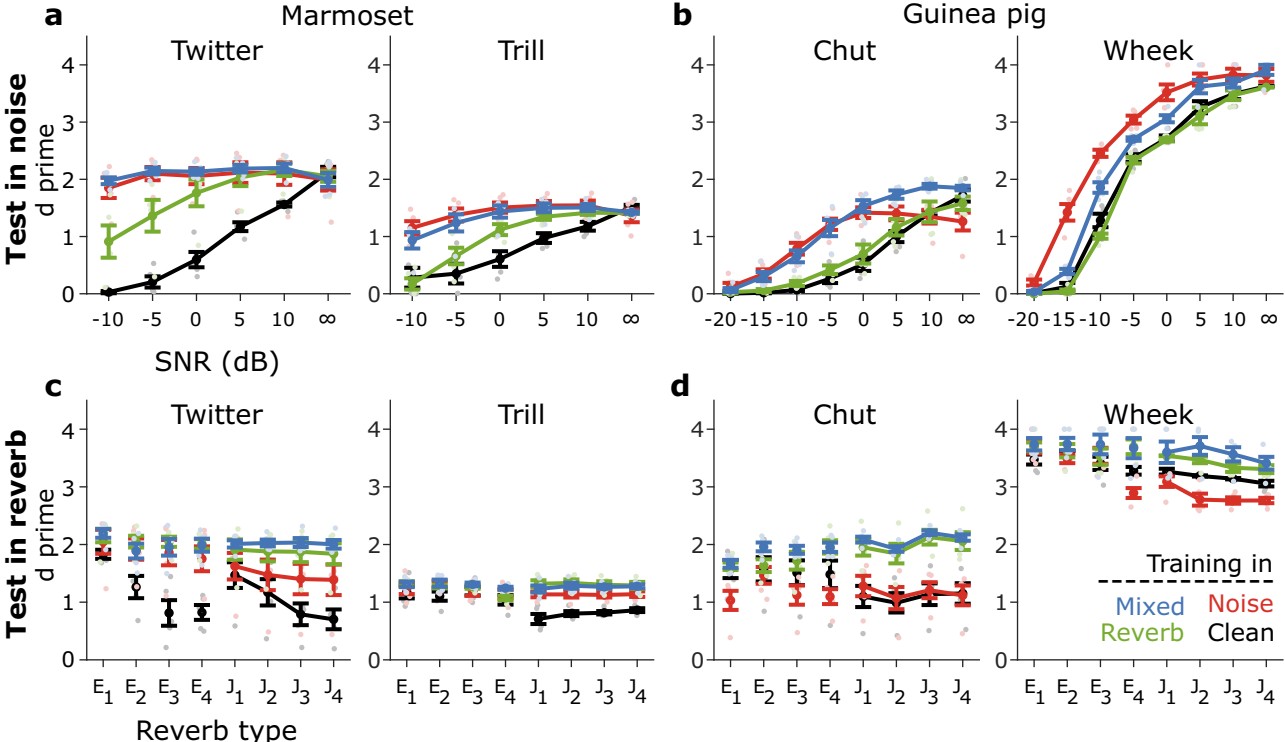

**Fig. 3 Condition-specific training improved performance but benefits were typically limited to the same condition. a, b** Model performance was quantified by the sensitivity metric, **d′**, in different levels of noise for several types of marmoset calls (**a**) and guinea pig calls (**b**). **c, d** Model performance for marmoset (**c**) and guinea pig (**d**) calls in reverberant conditions. The line colors correspond to the training condition: black—training with clean calls, red —training with noisy calls, green—training with reverberant calls, and blue—training in a mix of noisy and reverberant calls. The benefit was the largest for the model trained with mixed-condition calls and generally similar to the benefit when the testing condition matched the training condition. Lines correspond to means and error bars denote ±1 s.e.m. (*n* = 5 model instantiations). Statistics are reported in Supplementary Table S1. Symbols (dots) are slightly jittered along the *x*-axis to improve visibility.

vocalization categories early in development could conceivably occur in a range of different listening environments including clean, noisy, and reverberant conditions. But features learned during early exposure to a limited set of conditions would not be sufficient for all possible conditions that an individual might encounter over their lifetimes. An alternative approach would be to learn a small set of features in clean or a limited set of conditions and use other adaptive neural mechanisms to either modify noisy inputs to simulate clean inputs or modify feature properties to handle noisy inputs. As a first step towards modeling such adaptive mechanisms, we characterized the effects of additive white Gaussian noise or reverberation on the activity at the spectrotemporal input layer, which serves as the input to the FD layer. We also characterized the effects of noise or reverberation on the $V_m$ responses of FDs, which were trained in clean conditions.

**Noise and reverberation affected response distributions of both the spectrotemporal and feature-detector layers.** The distributions of response amplitudes across all frequency and time bins at the spectrotemporal layer, i.e., the FD inputs (Fig. 4b), and across all time bins of the FD $V_m$ responses (Fig. 4c) are shown in Fig. 4. Similar to the effects of adding noise to the acoustic waveform (i.e., increase in mean and decrease in standard deviation)[18], noise increased the mean [2.6 (clean), 3.0 (0 dB), and 3.5 (−10 dB)] and reduced the standard deviation [2. 7 (clean), 2.0 (0 dB), and 1.9 (−10 dB)] of the neural activity in the dense spectrotemporal layer (FD inputs, red and green lines in Fig. 4b). The FD $V_m$ responses also showed an increase in the mean amplitude (rightward shift in distributions, Fig. 4c) and a

decrease in its standard deviation (smaller spread in distributions, Fig. 4c) with increasing noise level. Similar effects on FD input and output probability distributions were also observed for reverberant conditions (Fig. 4f-h). Reverberation slightly increased the mean [2.63 (clean), 2.74 (T30 = 128 ms), 2.71 (T30 = 644 ms)] and substantially reduced the standard deviation [2.67 (clean), 2.23 (in both 128 ms and 644 ms T30 conditions)] at the FD input stage (Fig. 4g). However, what is relevant for categorization in our model is whether a given feature was detected, i.e., whether the maximum value of the FD $V_m$ response (triangles) exceeded the threshold of that FD (dashed lines). We observed that the maximal FD $V_m$ response in fact decreased with increasing noise/reverberation level. The maximum $V_m$ response was also lower in the two reverberant conditions (Fig. 4h). This is likely because the FD $V_m$ response is the extent to which a given feature 'matches' (using the max. cross-correlation value as a metric) the input, and the max. cross-correlation values progressively decreased with increasing noise level. In the section above, we had observed that features trained in noisy conditions exhibited lower thresholds than those trained in clean conditions. This decreased threshold likely compensates for the lower maximal values in the FD $V_m$ response distributions, thereby resulting in a supra-threshold FD output.

This observation of obtaining a supra-threshold FD output in noisy conditions by lowering the threshold in noise-trained features led us to explore two alternative strategies to counter these effects of noise/reverberation to improve model performance with a single set of FDs trained in the clean condition. First, we implemented contrast gain control, which normalizes the input to FDs such that the mean and standard deviation do

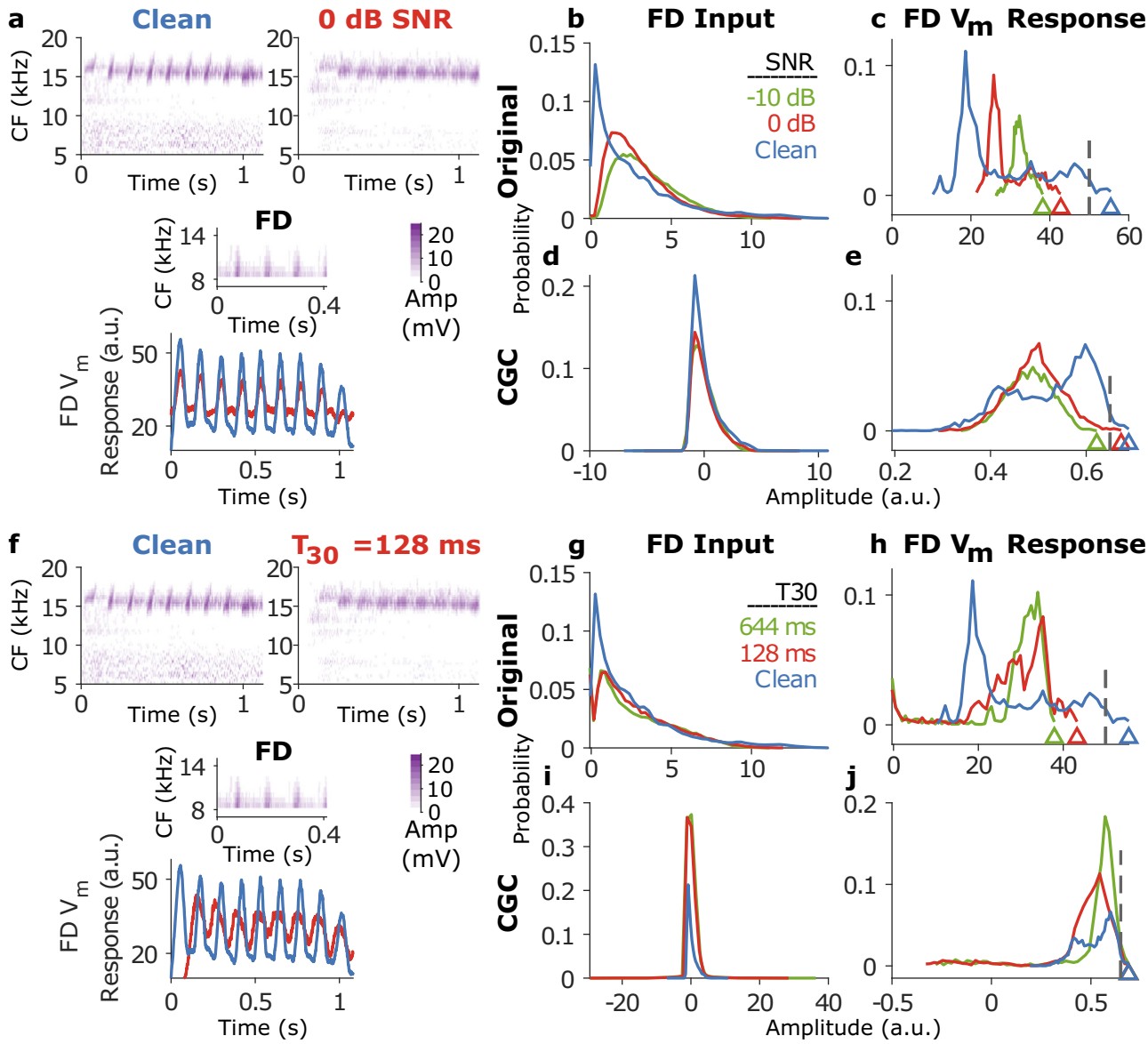

**Fig. 4 Noise and reverberation affected response distributions of the cochleagram as well as FD $V_m$ responses. a** Top, example cochleagrams for a marmoset twitter call in clean and 0 dB SNR conditions. middle, example cochleagram of a most informative feature for categorizing twitters. bottom, the $V_m$ response of the example FD had a lower standard deviation in the noisy condition (red) compared to the clean condition (blue). **b, c** Response distribution of the cochleagram (**b**) and response (**c**) of the FD in **a** (without contrast gain control) for three different SNRs. Dashed line in **c** represents the threshold of the FD learned during training. Triangles in **c** denote the maximum FD response value, which is the quantity subject to thresholding. **d, e** Same as **b–c** but with contrast gain control included in the model. **f** Top, the same cochleagram as in **a** [in clean (left) and reverberant (J1 reverberation condition with T30 = 128 ms)]. Middle, the same feature as in **a**. Bottom, the $V_m$ response of the feature detector had a slightly higher mean but substantially lower fluctuations in the reverberant condition compared to the clean condition. **g–j** Same format as **b–e** but for the reverberant condition in **f**. a.u. arbitrary units, CF characteristic frequency, CGC Contrast gain control, FD feature detector, SNR signal-to-noise ratio.

not change with noise/reverberation level (Fig. 4d, i). Note that this is an upper bound on stimulus contrast restoration (i.e., complete restoration within the receptive fields of FDs) as neurophysiological data show that the auditory system only partially restores stimulus contrast[19,34]. Second, because noise decreases the maximum value of the FD $V_m$ response (Fig. 4c, h), the threshold can be dynamically varied based on the level of noise/reverberation to improve feature detection in noise. In the following sections, we test the efficacy of these two adaptive mechanisms.

**Contrast gain control improved model performance in noise and in reverberation.** In the core model (i.e., the model without

contrast gain control), the FD STRF was demeaned and normalized to have a standard deviation of 1. This STRF was cross-correlated with the input cochleagram, and as such, there was no stimulus-dependent contrast gain adjustment to the input cochleagram. In the first adaptive addition to the model, we implemented a version of contrast gain control (without any top-down modulatory influences) to study its effects in isolation (note that combined effects are addressed at the end of the next section). To computationally implement contrast gain control, we demeaned both the input cochleagram (within the FD bandwidth) and the FD STRF and normalized both to have a standard deviation of 1. After this normalization, cochleagram (FD input) amplitudes were largely overlapping at all tested noise levels

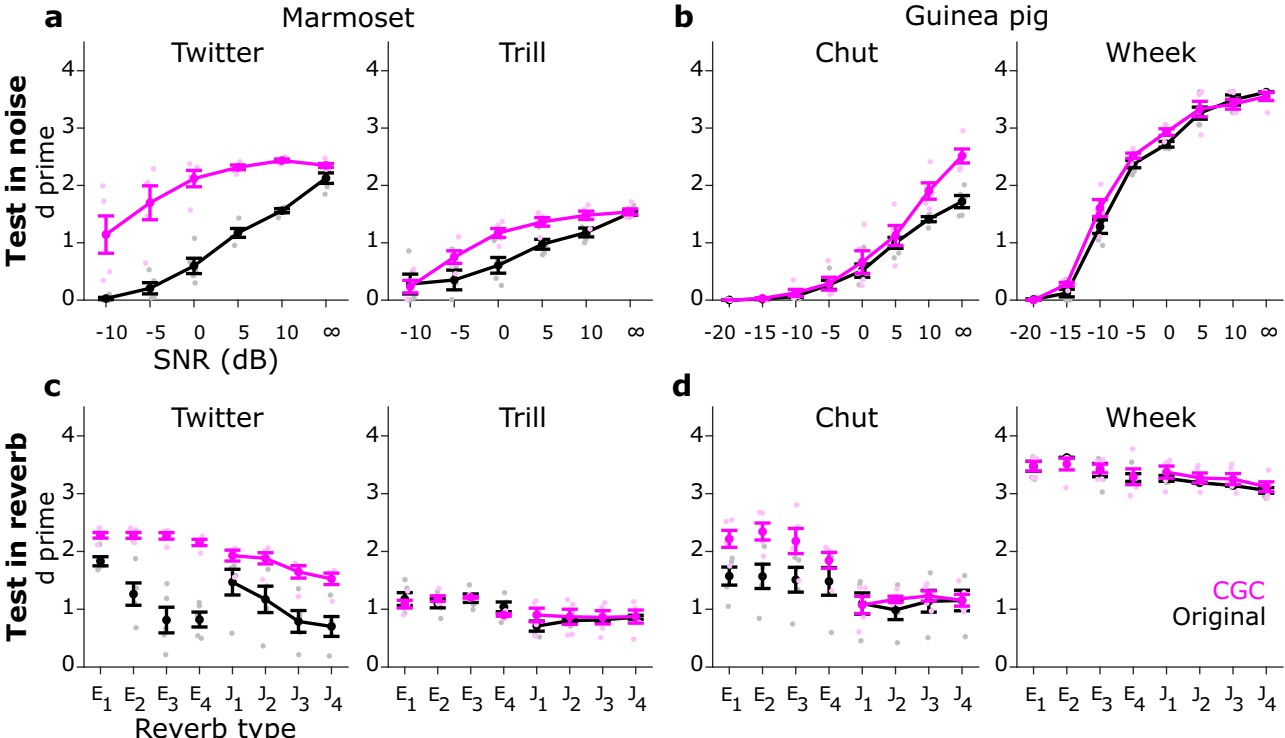

**Fig. 5 Contrast gain control improved model performance in noise and reverberation.** Same format as Fig. 3. **a, b** Model performance (d prime) in noise for marmoset (**a**) and guinea pig (**b**) call types. **c, d** Model performance in reverberation for marmoset (**c**) and guinea pig (**d**) call types. Black and magenta lines correspond to the model without and with contrast gain control, respectively. Lines correspond to means and error bars denote ±1 s.e.m. ($n = 5$ model instantiations). Statistics are reported in Supplementary Table S4. Symbols (dots) are slightly jittered along the x-axis to improve visibility.

(Fig. 4d) and reverberation conditions (Fig. 4i), and the range of FD $V_m$ responses was limited between −1 and 1. Compared to the model without gain control, the FD $V_m$ response distributions for the gain control model showed increased overlap, and the maximal value of the $V_m$ response distributions crossed the threshold at more adverse noise levels (Fig. 4e) and reverberation conditions (Fig. 4j). When we tested categorization performance, we saw that contrast gain control improved model performance both in noise and in reverberation for most call types (Fig. 5, Supplementary Table S4). However, the trends of these benefits were rather heterogenous (e.g., high improvement at low SNRs for Twitter but at high SNRs for Chut, and little benefit for Wheek in either noise or reverberation). Overall, these results demonstrated that contrast gain control could improve auditory categorization in noise and in reverberation, but even a perfect implementation of contrast gain control was insufficient to obtain significant benefits across all stimulus types.

**Model performance improved with top-down excitability modulation, but the magnitude of modulation scaled with SNR and not with reverberation strength.** Next, we evaluated the effect of top-down influences on model performance without contrast gain control being present. As mentioned earlier, noise reduced the maximum $V_m$ response value of a FD. But crucially, this reduction occurred for both within-class as well as outside-class calls for a FD (Fig. 6a). Therefore, even though both within-class and outside-class distributions of maximum FD $V_m$ responses were shifted down to lower values, these distributions retained some degree of separation, and optimal performance could theoretically be obtained by scaling down the FD threshold appropriately. We estimated the optimal threshold ratio for each condition (each SNR or reverberant condition) as the ratio value that maximized classification in that condition (quantified using

the mutual information between true and predicted call types). This optimal threshold ratio approximately linearly scaled with SNR (in dB) for both marmoset and guinea pig calls (Fig. 6c, d) and was more or less consistent across call types for each species. This was not the case for different reverberation conditions, however (Fig. 6e, f); in this case, the optimal threshold ratio was about the same across many tested reverberant conditions. In this implementation of the model, we used our knowledge of the stimulus SNR to implement this scaling. The brain would not have access to the stimulus SNR a priori, but as we show in the next section, stimulus SNR estimated from population activity at the dense stage in a biologically feasible manner can be used for excitability modulation with comparable performance benefits. Biologically, the reduction in threshold with noise level could be accomplished by increasing the excitability of FD neurons, i.e., by reducing the distance between each FD neuron's resting membrane potential and spike threshold. One possible implementation of excitability modulation in a neural circuit could be the disinhibition of FD neurons by cholinergic or noradrenergic inputs acting on canonical disinhibitory circuit motifs[25–30].

Next, we quantified model performance as a result of the top–down modulation of FDs. Model performance improved significantly and consistently for all call types in the SNR conditions (Fig. 7, Supplementary Table S4), and typically benefits scaled with the magnitude of top-down modulation (i.e., optimal ratio in Fig. 6c, d). Model performance improved but to a lesser extent for reverberant conditions (lower $\chi^2$ values in Supplementary Table S4). In summary, these results show that top-down modulation of FDs improves model performance, but these benefits are greater in noisy conditions than in reverberant conditions.

Finally, we also assessed the improvement in model performance when both contrast gain control and top-down

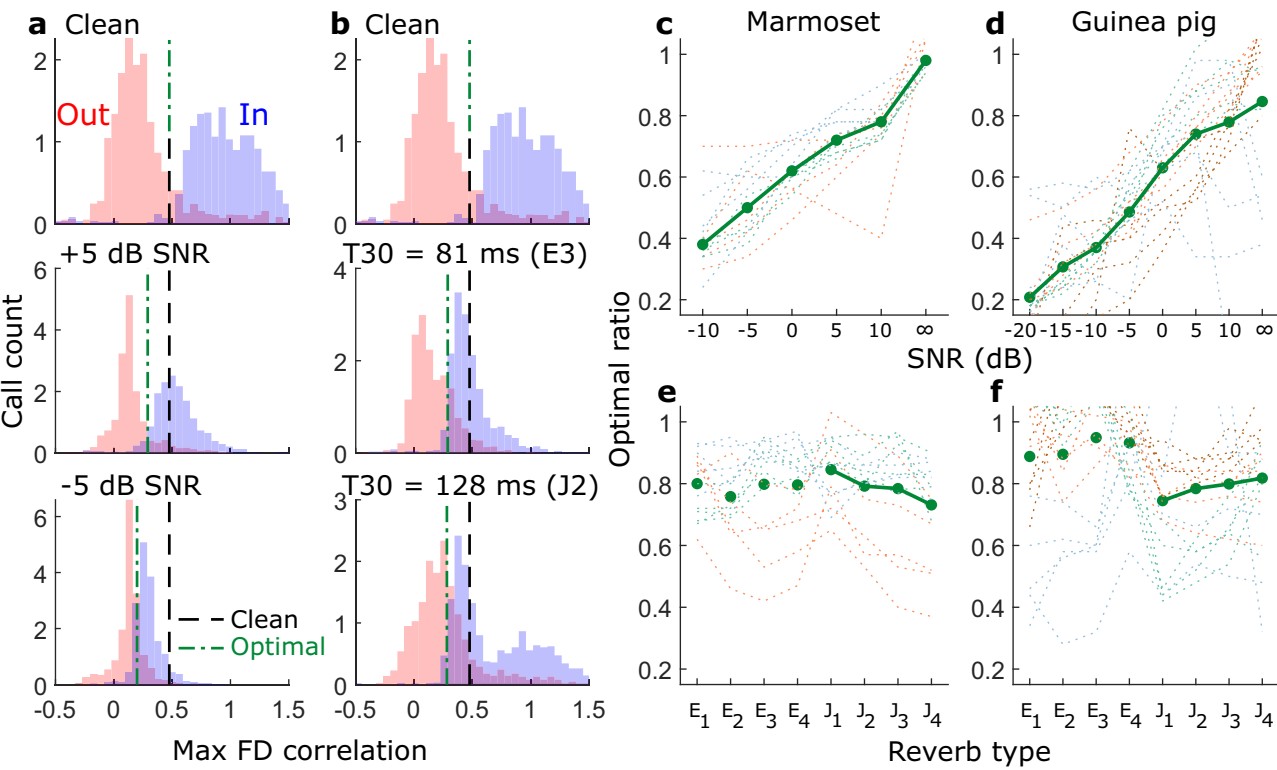

**Fig. 6 The distribution of feature-detector responses linearly scaled with SNR (in dB) but not with the strength of reverberation. a** Distributions of $V_m$ response maximum for a Twitter FD for 500 within-class (blue) and 500 outside-class (red) calls in clean (top), +5 dB SNR (middle), and −5 dB SNR (bottom) conditions. Black dashed line denotes the threshold learned during clean training; green dash-dotted line denotes the optimal threshold in each SNR condition. **b** Distribution of $V_m$ response maximum for the same Twitter FD as in **a** for 500 within-class (blue) and 500 outside-class (red) calls in clean (top), mild reverberation (middle), and moderate reverberation (bottom) conditions. **c, d** The optimal threshold values for different SNRs for marmoset **c** and guinea pig **d** call types. **e, f** The optimal threshold values for different reverberant conditions for marmoset (**e**) and guinea pig (**f**) call types. **c–f** Thin dotted lines represent individual call types, and solid green lines with symbols denote the median value across all call types.

modulation were present (gray lines in Fig. 7). The model performance was better (higher $\chi^2$ values in Supplementary Table S4) when both mechanisms were present compared to when only one mechanism was present (with the one exception being the SNR-test condition for guinea pig calls where top–down alone was better than the combination of both mechanisms). Taken together, these results suggest that the combination of contrast gain control or top–down modulation can further improve model performance beyond the improvement that either mechanism offers.

Next, we directly assessed how the sum of benefits (performance improvement over the core model) due to individual mechanisms (contrast gain control and top-down modulation) compared to the benefit when both mechanisms operated simultaneously. For both noisy (Fig. 8a) and reverberant (Fig. 8b) conditions, benefits due to individual mechanisms summed linearly when the d prime value was <1, but sublinearly at higher performance levels. This sublinear sum of benefits could indicate a ceiling effect in performance. In other words, when the model performance had been boosted by contrast gain control to a certain degree, additional top-down excitability modulation no longer provided as much of a benefit as it did when applied to the core model. But it is important to note, however, that the addition of top-down excitability modulation did not negatively impact the benefit provided by contrast gain control. No consistent pattern was apparent in the combined benefit provided by these mechanisms across call types and species in noisy conditions. For example, while a greater benefit at noisier SNRs was observed for guinea pig calls (Fig. 8c, blue lines), the opposite was true for

marmoset calls (Fig. 8c, red lines). For call types of both species, no clear trend was apparent for reverberant conditions either (Fig. 8d). Therefore, we next assessed whether the benefits due to the adaptive mechanisms depended on the baseline performance of the core model (i.e., core model performance without contrast gain control or top-down modulation). In both noisy (Fig. 8e) and reverberant (Fig. 8f) conditions, the benefit of incorporating adaptive mechanisms was little to none when the performance of the core model was already high (d′>3), further highlighting a ceiling effect. Adaptive mechanisms typically did not provide any benefit when the core model performance was very poor (d′<0.5), suggesting a performance floor had been reached at highly adverse conditions. It was at an intermediate baseline performance range (d′ between 0.5 and 3) where adaptive mechanisms provided the maximal benefit. This variability in benefit is likely due to the underlying variability in the spectrotemporal properties of target (within-class) call types and how well-separated the target call features are from non-target (outside-class) call features. For example, as discussed earlier, guinea pig wheek calls are likely tolerant to both noise (by virtue of being spectrally separated from other calls) and reverberation (by virtue of the information varying at a time scale much longer than reverberation time scales). Thus, high baseline model performance could be maintained over a large range of SNRs and reverberation strengths, with little room for improvement by adaptive mechanisms. In contrast, guinea pig rumble calls are susceptible to degradation by both noise (by virtue of their large spectral overlap with other call types) and reverberation (by virtue of rapid envelope modulations being a distinguishing

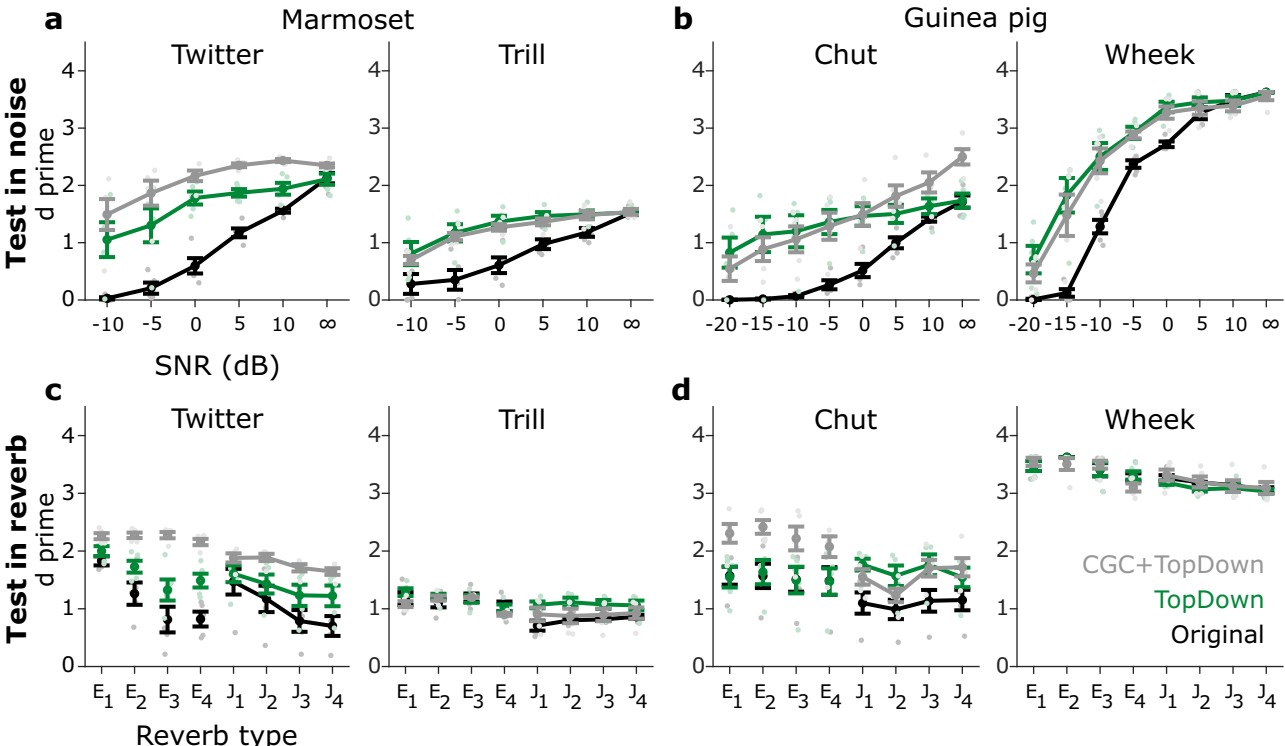

**Fig. 7 Top-down excitability control significantly improved model performance in noise and less so in reverberation.** Same format as Fig. 3.
**a**, **b** Model performance (d prime) in noise for marmoset (**a**) and guinea pig (**b**) call types. **c**, **d** Model performance in reverberation for marmoset (**c**) and guinea pig (**d**) call types. The line color corresponds to models with different mechanism: black –model without CGC or top-down modulation, green—model with only top-down modulation, and gray—model with both CGC and top-down modulation of FD excitability. Lines correspond to means and error bars denote ±1 s.e.m. ($n = 5$ model instantiations). Statistics are reported in Supplementary Table S4. Symbols (dots) are slightly jittered along the x-axis to improve visibility. Optimal threshold ratios in Fig. 6c, d were used to implement top-down modulation across conditions.

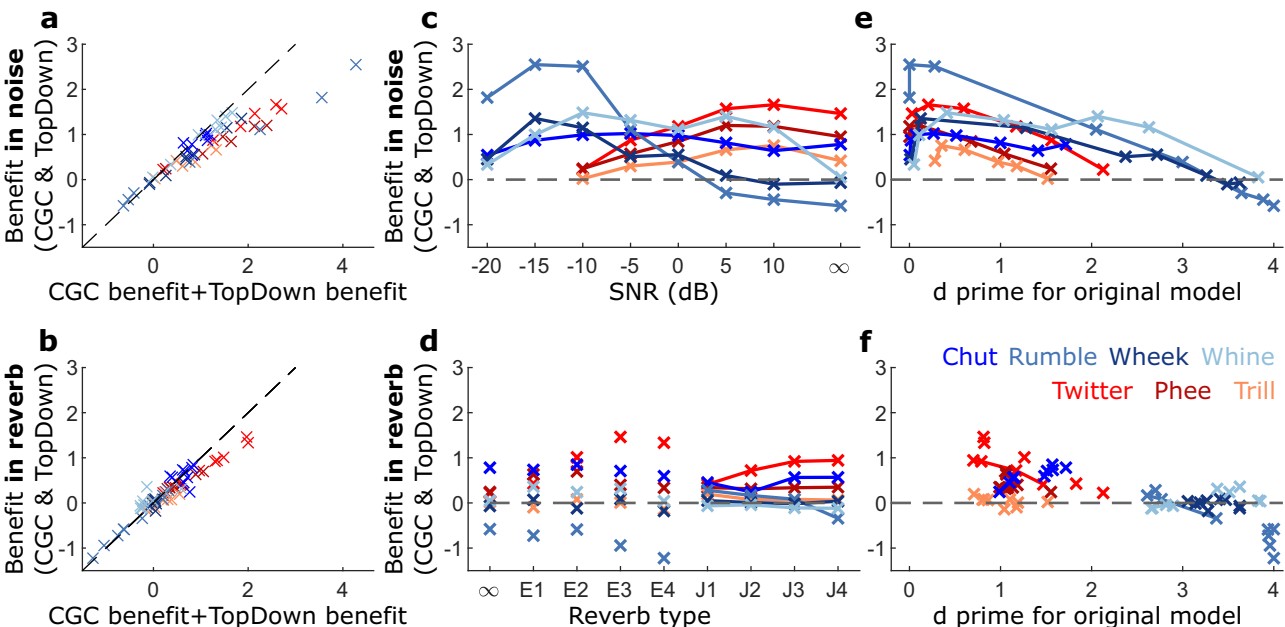

**Fig. 8 The performance of the original model predicts benefits due to contrast gain control and top-down modulation. a**, **b** The benefit when both adaptive mechanisms are operational (y-axis) versus the sum of benefits when either mechanism is operational in isolation (x-axis) in noisy (**a**) and reverberant (**b**) conditions. Marmoset and guinea pig call types are denoted by red and blue symbols; different shades correspond to different call types. The dashed gray line represents a line with a unity slope. **c**, **d** Benefit (in units of d-prime) when both mechanisms are simultaneously active as a function of SNR (**c**) and reverberation condition type (**d**). **e**, **f** Benefit when both mechanisms are simultaneously active versus baseline performance (i.e., the performance of the core model without any adaptive mechanism) in noisy (**e**) and reverberant (**f**) conditions.

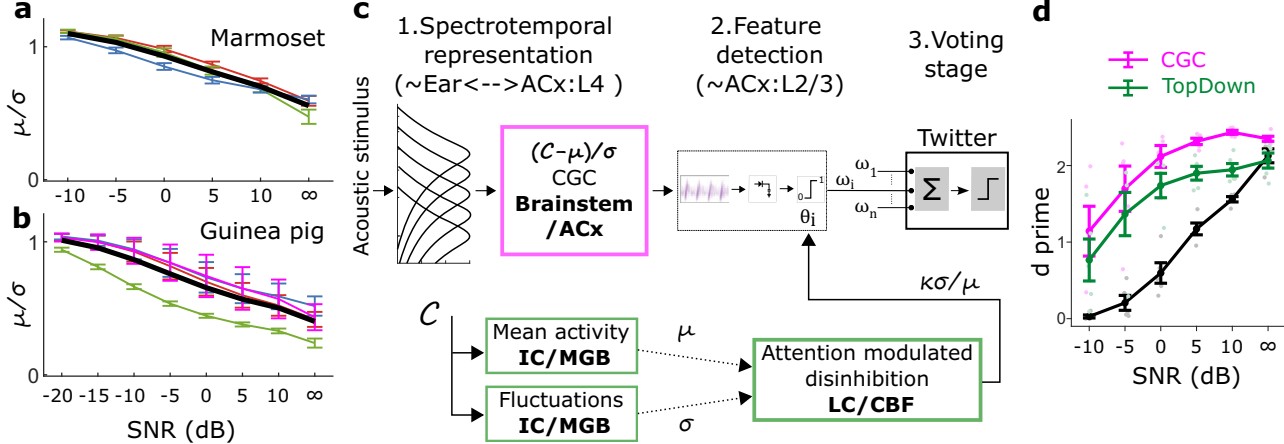

**Fig. 9 Biologically feasible implementation of contrast gain and top-down control for noisy conditions. a, b** The ratio of the mean ($\mu$) to standard deviation ($\sigma$) of the overall cochleagram at different SNRs for marmoset (**a**) and guinea pig (**b**) call types. Colored thin lines represent the mean (±1 standard deviation) for $\mu/\sigma$ for individual call types [marmoset calls in **a**: $n = 1055$ (twitter), 1090 (phee), 1080 (trill); guinea pig calls in **b**: $n = 432$ (chut), 253 (rumble), wheek (372), and 457 (whine)]; thick black lines indicate the median across calls of all types. **c** Contrast gain control (magenta box) can be implemented by demeaning the cochleagram and normalizing it by its standard deviation. Top-down control (green boxes) can be implemented by scaling the feature-detector threshold by a factor proportional to the ratio of the standard deviation to the mean of the cochleagram. **d** Model performance to classify Twitter calls from other call types was significantly better using biologically feasible contrast gain control $[\chi^2(2) = 70.1, p = 5.8 \times 10^{-16}]$ as well as top-down modulation $[\chi^2(2) = 36.6, p = 1.4 \times 10^{-9}]$ compared to the original model. Symbols (dots) are slightly jittered along the x-axis to improve visibility. ACx auditory cortex, CBF Cholinergic basal forebrain, CGC Contrast gain control, IC Inferior colliculus, LC locus coeruleus, MGB medial geniculate body, SNR signal-to-noise ratio.

feature). For this call type, adaptive mechanisms could improve the performance of the model at multiple SNRs and reverberation strengths. Overall, these results reveal that the high variability in the spectrotemporal structures of calls (necessary for separating call categories) could also lead to high variability in their susceptibility to different environmental degradations, thus requiring the presence of multiple adaptive mechanisms to cope with these challenges.

**Biologically feasible implementation of contrast gain control and top-down modulation of excitability.** The adaptive model implementations presented so far are engineering solutions—for example, we implemented perfect contrast gain control by measuring the response distributions of the spectrotemporal layer and normalizing the responses using the mean and standard deviation. However, such implementations may not be biologically realistic. For example, contrast gain control in the auditory system partially restores stimulus contrast at a neuronal level over multiple processing stages, whereas our contrast gain control implementation aimed to restore contrast completely for each FD at a single stage. To determine the extent to which biologically feasible implementations of the mechanisms presented above aid model performance in noise, we explored the following alternate implementations (Fig. 9). For contrast gain control, instead of modifying cochleagrams (zero mean and unity variance) in a boxcar manner (matched to the bandwidth and duration of the FD) for each time step of the cross-correlation operation, we subtracted the mean of the entire cochleagram and normalized it to have unity variance. This way, contrast restoration was not optimized to within the bandwidth and duration of individual FDs. Such an operation could be performed in neural circuits by widely tuned inhibitory neurons that have been implicated in contrast gain control in the visual system[35,36], although the auditory system may have different implementations[37]. We retrained the model with this cochleagram normalization to learn FDs. Similar to the previous model with contrast gain control, this modified implementation of contrast gain control also led to

significantly better performance than the original model (i.e., without any contrast gain control, Fig. 9d).

To implement top-down control, we had previously used knowledge of stimulus SNR, which is available to the experimenter but is not accessible to biological neural circuits. Therefore, we first asked whether the mean ($\mu$) and standard deviation ($\sigma$) of cochleagrams can be used to estimate the acoustic SNR. The mean can be estimated from the population activity of neurons at multiple stages in the auditory pathway. The standard deviation (or modulation) can be estimated biologically by neurons in the inferior colliculus, whose responses have been shown to be sensitive to temporal modulations[38–40]. We found the ratio of $\mu$ over $\sigma$ to be (negatively) correlated with the acoustic SNR (Fig. 9a, b). Next, we learned the optimal threshold ratio ($\kappa$) based on the $\mu/\sigma$ of cochleagrams, thus doing away with the need to use the acoustic SNR. A single $\kappa$ (median across all calls) was estimated for all call types of each species (marmoset or guinea pig). To demonstrate feasibility, for a test call type (marmoset Twitter call), we used $\kappa$ to scale the threshold of FDs for individual cochleagrams. Similar to the top-down model, the performance of this biologically feasible model was significantly better than the original model (Fig. 9d).

**Comparison of model performance with behavior.** Finally, we compared the performance of the proposed models with guinea pig behavioral performance (Fig. 10). Guinea pigs were trained on a go/no-go task using only clean calls, where the target/distractor call pair was either chut/rumble or wheek/whine[7]. Guinea pig call categorization was then tested in noisy conditions at different SNRs and in different simulated reverberant environments. Note that guinea pig behavior is governed by two underlying processes —the recognition of the correct call category and the expression of that recognition by performing an operant action. The model only captures the former process, and while other factors such as motivation, attention, and past performance might influence guinea pigs' operant behavior, the model perfectly reports the recognized call category. Thus, the models are expected to over-

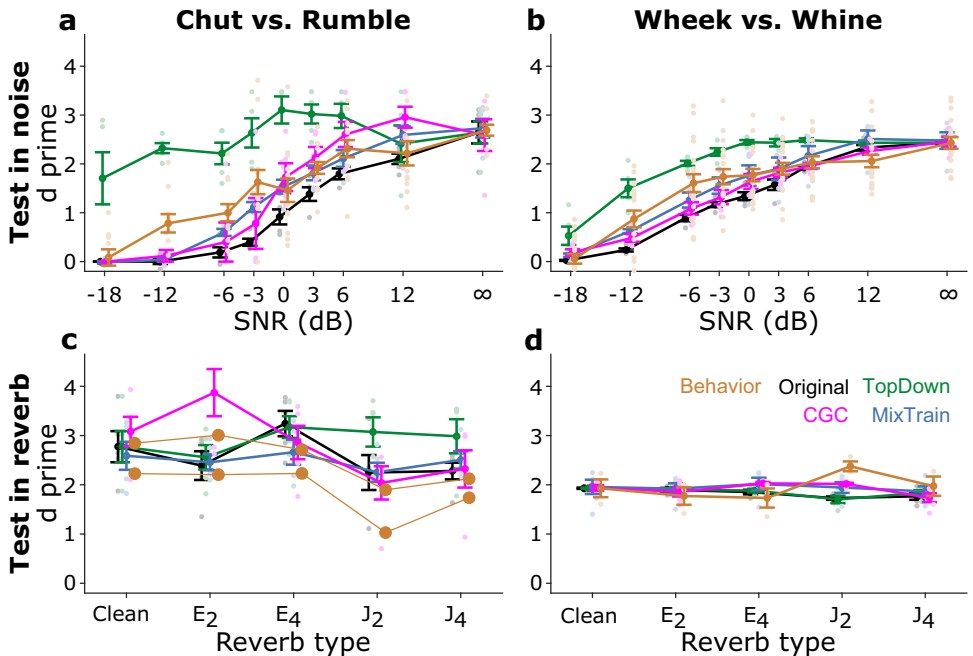

**Fig. 10 Comparing the performance of various models with the behavioral performance of guinea pigs. a**, **b** Performance of various models and guinea pigs (behavior) for two different go/no-go tasks at several noise levels. Guinea pigs were trained on a go/no-go task, where the target/distractor pair was either chut (go)/rumble (**a**) or wheek (go)/whine (**b**). The performance metric for guinea pig behavior was determined as d-prime = norminv(hit rate) −norminv(false alarm), where norminv is the inverse Normal cumulative distribution function. Model d-prime was first estimated similar to guinea pig behavior, but then transformed using a sigmoidal function that was learned to match model and guinea pig performances in the clean condition (Supplementary Figure S4, also see Methods). **c**, **d** Model and guinea pig (different set of animals compared to **a**, **b**) performances (behavior) for the same two go/no-go call pairs as in **a**, **b** when presented in various reverberant conditions. Model performances were also scaled (similar to noise conditions) to match model and guinea pig performances in the clean condition. Lines correspond to means and error bars denote ±1 s.e.m. $n = 5$ model instantiations in **a**–**d**; $n = 4$ (**a**, **b**), 2 (**c**), and 3 (**d**) guinea pigs. As $n = 2$ for behavior data in **c**, individual lines are shown instead of error bars. Symbols (dots) are slightly jittered along the x-axis to improve visibility.

perform when compared to guinea pig behavior. To adjust for this over-performance, we learned an additional mapping to match the model and guinea pig performances in the clean condition (overlapping points for behavior and models at point marked '∞' in Fig. 10a, b and 'Clean' in Fig. 10c, d). This mapping, learned only using clean calls, was then applied to adjust model performance in the noisy and reverberant conditions (see Methods). The adjusted performance of the original model without adaptive mechanisms (black lines, Fig. 10a, b) was consistently worse than guinea pig behavioral performance (orange lines, Fig. 10a, b) in noisy conditions. Both mixed training and contrast gain control improved adjusted model performance at adverse SNRs, but were insufficient by themselves to match guinea pig behavioral performance (blue and magenta lines, Fig. 10a, b). In contrast, optimal top-down excitability modulation improved adjust model performance to levels better than guinea pig behavior (green lines, Fig. 10a, b). Adjusted model performance for the original as well as extended models broadly matched guinea pig behavior in the reverberation conditions, and the models as well as behavior did not show a strong dependence on reverberation strength (Fig. 10c, d). Taken together, these results suggest that a combination of adaptive mechanisms is necessary for the hierarchical model to recapitulate guinea pig behavioral performance for call categorization.

## Discussion
In this study, we explored biologically realistic extensions to a hierarchical model for auditory categorization to improve its performance in realistic listening conditions. We tested three mechanisms including task-specific training, contrast gain

control, and top-down modulation of FD sensitivity. All three mechanisms improved model performance when tested in noisy and reverberant listening environments; however, the trends and magnitude of improvement varied across mechanisms. We demonstrated that the proposed contrast gain control and top-down modulation mechanisms can be implemented by biological circuits. Finally, the comparison of model performance with guinea pig behavioral performance revealed that these mechanisms could be employed in a flexible manner to accomplish call categorization based on both task demands and call spectro-temporal properties.

**Task-specific training led to little systematic differences in the spectrotemporal properties of feature detectors.** Consistent with previous studies, our results showed that task-specific training can improve auditory categorization performance in noisy and in reverberant conditions. The benefit was typically greater at more adverse conditions (e.g., between −10- and 0-dB SNR, and for J1-J4, which were the strongest reverberant conditions). Interestingly, the noise-trained model frequently showed non-monotonic performance curves as a function of SNR and performed worse in clean conditions than models trained on clean calls (e.g., Twitter, Chut). While a small amount of noise during training can have a regularization effect to improve the generalizability of models[13], uniformly using all SNRs during training likely led to overfitting of the model to noisy conditions. Overfitting may be avoided by appropriately weighing SNRs during training by following SNRs that animals and humans naturally experience.

Even though task-specific training improved model performance in the same task, the benefits were significantly reduced or

sometimes negative when training and testing conditions were different. Mixed training, i.e., training in multiple types of degraded listening conditions, however, could ameliorate these generalization issues. Task-specific training did not lead to any systematic changes in the spectrotemporal properties of FDs (except for duration for noise-trained models for marmoset calls). But although training in adverse conditions can help in real-world categorization, the listening conditions that are present during category acquisition may not encompass all the different listening conditions an organism might encounter in its lifetime. Thus, adaptive mechanisms operating after training that can cope with novel listening environments might present a more general solution.

**Contrast gain control improved model performance, but benefits varied across call types.** Contrast gain control is ubiquitous in both visual and auditory pathways, and more specifically, it has been reported along the auditory pathway from the auditory nerve[19] to the auditory cortex[17]. Contrast gain control can tune the dynamic range of individual neurons to the statistics of the incoming sound, which can aid in adverse listening conditions. For example, additive noise increases the mean power of the signal while reducing its variance[34]. The effect of reverberation on communication signals such as speech and vocalization is also similar as reverberation smears the temporal envelope, thus reducing the contrast in the signal[41]. Contrast gain control significantly improved model performance both in noise and in reverberation. However, the trends of improvement were quite heterogeneous across call types, suggesting that the benefits of contrast gain control depend on the spectrotemporal properties of the signal. Future studies are needed to understand what features determine the benefit due to contrast gain control.

While contrast gain control has been documented along several regions of the auditory pathways[16–22], the underlying neural circuits are not settled. Unlike the visual system, neither shunting inhibition by parvalbumin-expressing interneurons[35,42] nor contrast-dependent variability in membrane potential responses[43] seem to contribute to contrast gain control in the mouse auditory cortex[37]. A promising alternate mechanism that may contribute to contrast gain control is synaptic depression, by which neurons adapt to steady backgrounds (e.g., in noisy or reverberant conditions), thus maintaining their dynamic range to respond to transient signals (e.g., onset of the next syllable in ongoing speech)[44–46]. Explicitly incorporating synaptic depression in our model represents one way in which the biological feasibility of our model can be further increased, and by which additional testable predictions can be generated.

**Top-down modulation led to divergent benefits in noise and in reverberation.** Attention plays a critical role in shaping perception in challenging environments across modalities[47]. Effects of attention are thought to be mediated by top-down modulation, which can shape information representation along the auditory pathway[23,24]. We modeled such top-down feedback control by scaling the excitability of FDs to optimally improve task performance in noise and in reverberation. Interestingly, the optimal scale was proportional to noise level (in dB), but it was nearly constant across different reverberant conditions. Similarly, benefits of top–down control were greater in noisy conditions than in reverberant conditions (as indicated by $\chi^2$ values in Supplementary Table S4). One interpretation of this result is that increased listening effort is not beneficial in reverberant environments. While unexpected, this interpretation is supported by psychoacoustic studies in humans that demonstrate that listening effort (as indexed by the pupil diameter, which correlates with cholinergic activation[48,49]) scales with noise level but not with the strength of reverberation[50–52]. Our model cannot explain at present, however, why FD excitability does not scale with reverberation strength. One possible explanation is that the time scale of information that provides contrast between call categories is much greater than the reverberation T30s used in this study. These results suggest that top-down attentional mechanisms are primarily beneficial in noisy environments, but these benefits are substantially reduced in reverberant environments.

Top-down processes, such as attention or listening effort, could modulate auditory cortical activity through several proposed pathways[23,24]. One example of such a mechanism involves a canonical microcircuit in which vasoactive intestinal polypeptide (VIP)-expressing interneurons disinhibit excitatory cortical principal cells by inhibiting somatostatin and parvalbumin-expressing interneurons (which inhibit principal cells). Since VIP-expressing interneurons are activated by cholinergic inputs (at least in the primary visual cortex), this disinhibition could be recruited during effortful listening. Two other relevant pathways that also modulate cortical excitability involve the noradrenergic locus coeruleus[25,26] and the cholinergic basal forebrain[27–30]. While both these pathways generally modulate auditory cortical activity globally (i.e., in a frequency-independent manner), which is similar to our implementation of top-down modulation (e.g., FDs for all calls were similarly modulated at the same noise level or reverberation condition), it remains to be seen whether other top-down mechanisms exist that are optimized to the spectrotemporal properties of the incoming sound. Further experiments are necessary to directly test whether FD excitability can be modulated via one of these proposed pathways.

In conclusion, we have extended a versatile auditory categorization model, whose strengths include its straightforward biological interpretability and efficient trainability, modularity to include adaptive neural mechanisms, and showed that mechanisms such as contrast gain control and top-down feedback control can improve auditory categorization performance in challenging listening environments.

## Methods
All procedures followed the NIH Guide for the Care and Use of Laboratory Animals and were approved by the institutional animal care and use committee of the University of Pittsburgh (protocol #21069431).

**Stimuli.** The model was trained to perform an auditory categorization task in which it discriminated one call type from other conspecific call types. Two sets of vocalization stimuli were used, including calls from marmosets and guinea pigs, two highly vocal and social species. Marmoset calls have been described in detail in a previous study[32] and have been used in the previous version of the model[5]. Briefly, these calls were recorded from eight adult marmosets of either sex living in a marmoset colony using an array of directional microphones. Guinea pig calls were primarily recorded from four male and one female guinea pigs. Male guinea pigs were placed in pairs in a sound-attenuating booth, sometimes in two different chambers separated by an acrylic divider[33]. A directional condenser microphone, suspended from the sound booth ceiling, was used to record these vocalizations. To record wheek calls, the microphone was placed outside the guinea pig cages in the colony using a tripod. Guinea pig calls were recorded using Sound Analysis Pro 2011, sampled at 48 kHz, and manually curated using Praat[53].

Noisy calls were generated by adding white Gaussian noise to calls. Reverberant calls were generated using eight different impulse responses (Supplementary Figure S1). The strength of reverberation was quantified by the T30 metric, which indicates the duration for signal energy to decay to 30 dB below the original value (Supplementary Figure S1). Four of these impulse responses (denoted by J1-J4) have been previously used for human speech perception studies[54]. These impulse responses were originally generated using Odean[55] and had the following T30s—128 (J1), 236 (J2), 461 (J3), and 644 (J4) ms. The other four impulse responses were downloaded from online sources (Open Acoustic Impulse Response library [Open AIR], www.openair.hosted.york.ac.uk, and other now-defunct websites) and corresponded to the following naturalistic environments—snow site (E1, T30 = 7 ms), plastic bin (E2, 57 ms), living room (E3, 81 ms), and forest (E4, 124 ms).

**Model architecture**. The model used in this study was similar to the model we have previously used[5,7]. Briefly, the model was hierarchical and comprised three stages. The first stage was a spectrotemporal stage that used a phenomenological model[6] of the auditory periphery to generate the cochleagram, which is a physiologically accurate spectrotemporal representation of the stimulus. Inner hair cell voltage was used as the response instead of the auditory nerve fiber firing rate (which was used in our previous publication[5]) to achieve faster implementation. Cochleagrams were generated at a sampling frequency of 1 kHz and the center frequencies spanned 200 Hz to 20 kHz in 0.1-octave steps. Marmoset calls were processed using a marmoset head-related transfer function[56], which was estimated using the GRABIT.m MATLAB function[57]. Cochlear tuning in the periphery model was set to "human" for marmoset cochleagrams and to "cat" for guinea pig cochleagrams. The overall stimulus level of clean and degraded calls was set to 65 dB SPL. In the second stage, the cochleagram was filtered by the STRF of a feature detector (FD) to obtain its membrane potential response or $V_m$ response, following which a threshold was applied to the maximum of the filter output to obtain a binary output (1 if maximum $V_m$ response >threshold). Finally, outputs of a set of maximally informative FDs were weighted and combined in a voting stage to obtain the final response of the model for a call. The set of maximally informative FDs as well as their thresholds and weights were learned during training, as described next.

The model was trained to classify one call type (within-class) from all other conspecific call types (outside-class). Marmoset models were trained using 500 within-class calls and 500 outside-class calls, and tested with a non-overlapping set of within- and outside-class calls (500 calls each). Guinea pig models used 70% of all available calls for training and the remaining for testing. Therefore, the number of calls for training (and testing) was different for different call types (number of training calls for chut=248, rumble=176, wheek=230, whine=300). During training, initial features were generated by randomly segmenting rectangular blocks (i.e., random center frequency, bandwidth, onset time, and duration) of within-class cochleagrams. The number of initial features was 6000 for marmoset call types and 4000 for guinea pig call types.

For each FD, its spectrotemporal pattern was used as the STRF to filter a cochleagram, and the maximum value of the $V_m$ response was used to construct distributions for within-class and outside-class calls. The threshold of the FD was set to the correlation value that maximized the mutual information between the binary output (after applying the threshold on the FD $V_m$ response) and the stimulus category (within-class and outside-class). The weight of the FD was set to the log-likelihood ratio of this classification. After estimating the threshold and weight of individual FDs, a greedy search was implemented to estimate a set of maximally informative and least redundant FDs (MIF set) for each model. This was an iterative process, where we sequentially added FDs to the MIF set to increase the hit rate without increasing the false alarm rate. The maximum number of FDs in an MIF set was set to 20. We trained five instantiations of the model for each call type (by using non-overlapping feature detectors) to assess the reliability of model performance statistically. Model performance was quantified using receiver-operating curve (ROC) analysis; we first estimated the area under the ROC curve (AUC) using model output (i.e., the output of the voting stage) for within-class and outside-class calls and then estimated the sensitivity index (d') from this area as: $d' = \sqrt{2} \times \mathrm{norminv}(AUC)$, where norminv is the inverse of the Normal cumulative distribution function. The same set of within-class and outside-class calls were used to quantify performance in clean, noisy, and reverberant conditions.

**Feature-detector properties**. We characterized FDs by estimating several properties such as threshold, center frequency, bandwidth, and duration. In addition, we also estimated the complexity (tailedness) of FDs using reduced kurtosis (i.e., 3 subtracted from the kurtosis, where kurtosis is the ratio of the fourth central moment to the fourth power of the standard deviation). Higher (lower) values for reduced kurtosis indicate a heavy-tailed (light-tailed) distribution relative to the Normal distribution. Distributions were plotted using standard violin plots[58] that show the median, the interquartile range, and the 1.5× interquartile range.

**Animal behavior**. Detailed protocols for guinea pig (GP) behavioral experiments have been previously described[7]. Briefly, food-restricted animals were trained on a go/no-go task, where the target(go)/distractor(no-go) stimulus pair was either chut/rumble or wheek/whine. All behavioral experiments were conducted in a custom-built sound-attenuating booth. The booth was virtually divided into 'home' and 'reward' regions. Sounds were presented from an overhead speaker, and the animal's position was tracked using an overhead camera. A trial was initiated when the GP entered the home region, and a stimulus (target or distractor) was presented after 3-5 s (randomly drawn from a uniform distribution). Following stimulus presentation, the animal either stayed in the home region ("no-go" response) or moved to the reward region ("go" response) within a 5-s window. The outcome of each trial was one of the following: hit (target stimulus and go response), miss (target stimulus and no-go response), correct rejection (distractor stimulus and no-go response), and false alarm (distractor stimulus and go response). Animals received a food pellet as a reward for each hit, but an air puff and a brief timeout (with lights off) for false alarms. Animals were first trained on only clean calls until their performance d-prime reached 1.5, and then tested in a different set of calls that were clean, or degraded by noise or reverberation. Animal training and testing were performed in a block design, where each block contained

up to five sessions with 40 trials in each session. For testing in noise, each block corresponded to a unique SNR. For reverberation testing, each block had a mix of clean and all reverberant calls. Behavioral data reported here were acquired from four animals for SNR conditions and two animals for reverberant conditions.

**Comparison between model and behavioral performances**. To compare the model performance with animal behavioral performance in the go/no-go task, we added a winner-take-all (WTA) stage at the back end of the model[7]. Briefly, for any stimulus (either a target or a distractor), we computed the normalized target model response and the normalized distractor model response. The normalized target (distractor) model response is calculated as the sum of the weights of all detected target (distractor) features divided by the sum of all feature weights for the target (distractor) model. Thus, the normalized response for either model ranged between 0 (no feature detected) and 1 (all features detected). The difference between the normalized target and distractor model responses was the WTA-stage output, which ranged between −1 (all distractor model features and none of the target model features were detected) and 1 (all target model features and none of the distractor model features were detected). For an ideal-observer model, a decision regarding whether the stimulus was a target or distractor could be made simply based on the sign of the above quantity (positive values—target; negative values—distractor). However, because the animal's decision could be additionally influenced by unmodeled variables such as motivation, arousal, and past performance, the ideal-observer model is expected to over-perform compared to animals.

To adjust our model to incorporate these factors, we learned a mapping to transform WTA output to decision probability such that the model performance (d') and behavioral performance matched in the clean condition. We then used the same mapping to transform the ideal-observer d' to adjusted d' in noisy and reverberant conditions. The mapping was based on a two-parameter (the slope $k$, and symmetric minimum and maximum $\lambda$) logistic function, $F(x; k, \lambda)$, where $F(x; k, \lambda) = \lambda + (1 - 2\lambda)/(1 + e^{-kx})$, which mapped WTA output to model go-rate for a stimulus. The d' metric was estimated in the same way as that for guinea pig behavior; for each call type, the go-rate was averaged across all target and distractor calls to estimate hit rate and false alarm rate. Then d' was estimated as d' = norminv(hit)—norminv(false alarm). A single set of $k$ and $\lambda$ were learned for five different instantiations of individual mechanisms for each target-distractor call type pair. $k$ and $\lambda$ were learned separately for testing in noise (Supplementary Figure S4b) and testing in reverberation (Supplementary Figure S4e) because different sets of animals were used in those behavioral experiments.

**Statistics and reproducibility**. All statistical analyses were performed in R[59] (version 4.2.1). The effects of various mechanisms were evaluated by constructing linear mixed-effect models (lme4 package[60]) and comparing model fits using anova (stats package). To evaluate the effect of a mechanism (six mechanisms in total: noisy training, reverberation training, mixed training, contrast gain control, top-down modulation, or contrast-gain control + top-down modulation) on the performance in a testing condition (i.e., in noise or in reverberation), two models were constructed (separately for each species):

1. full model: *dprime ~ testing_parameter + call_type\*mechanism + (1 | model_id)*
2. null model: *dprime ~ testing_parameter + call_type + (1 | model_id)*

where *testing_parameter* was either SNR in dB (interval scale) or reverberation type (nominal scale), *call_type* (nominal scale) was the conspecific call types used for each species, and *model_ID* (a random effect, nominal scale) was the model instantiation index. The term *call_type* was excluded for statistical analysis in Fig. 9 as the data were for only a single marmoset call type (i.e., twitter). Five separate model instantiations (with non-overlapping MIF sets) were trained for each call type to assess training convergence and reproducibility of our results.

The effect of training on model FD parameters (e.g., duration, CF) was quantified using the F-test as well as the partial eta squared[61] ($\eta_P^2$, estimated using the etaSquared function in the lsr package[62] with type II sum of squares) values, which approximates the fraction of total variance captured by training. A linear model (lm in stats package) was fit for each combination of species, training condition, and FD property with FD property value as the outcome. Predictors included call type, training condition, and their interaction. Center frequency was log-transformed.

**Reporting summary**. Further information on research design is available in the Nature Portfolio Reporting Summary linked to this article.

## Data availability
All guinea pig vocalizations used in this study are available at the following website: https://github.com/vatsunlab/CaviaVOX.

## Code availability
The code for the models is available at the following website: https://github.com/vatsunlab/Feature_based_auditory_model.

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

## Acknowledgements

This work was supported by funding from the NIH to S.S. (R01DC017141). The authors would like to acknowledge Dr. Xiaoqin Wang (Johns Hopkins University) for providing marmoset calls used for modeling. The authors are grateful to Manaswini Kar, Dr.

Marianny Pernia, and Kayla Williams for the guinea pig behavioral data. This research was supported in part by the University of Pittsburgh Center for Research Computing through the resources provided.

## Author contributions

All authors contributed to the conceptual development of the model. S.P. implemented all simulations and statistical analyses, building upon a preliminary implementation by S.T.L. S.S. provided overall guidance, supervision, and funding. S.P. wrote the manuscript with inputs from S.S.

## Competing interests

The authors declare no competing interests.
