## [Peer Review File · Communications Biology]

Reviewers' comments:

Reviewer #1 (Remarks to the Author):

* == Summary ==

The authors consider a model of robust auditory categorization. In particular, they extend a previous model to provide robustness to variability from the external environment, such as due to noise and signal reverberation. The authors consider three different model extensions, and compare each with task behavior observed in guinea pigs. The authors claim that models with adaptive mechanisms show better categorization, and produced responses most similar to the guinea pig responses, and thereby conclude that such adaptive mechanisms are important for robust auditory categorization.

Each of the three model extensions resulted in improved categorization, but the improvements were greater for some vocalization types, and some noise types than others, and this pattern of improvement varied substantially across the three different extensions.

Specifically:

A. For the condition-specific training, training in noise aided recognition in noise, training in reverberation aided recognition in reverberation. There was little evidence of a generalized benefit from either. Also, the authors note this is an expensive solution. New features must be learned and remembered for each noisy environment.

B. For the contrast gain control, in which the signal was de-measured and variance-normalized before feature detection, performance improved for both recognition of noisy and reverberant calls. However, very different improvements were observed for different call types. For example, "twitter"-detection was much improved but "wheek"-detection was not.

C. For the "top-down" mediated feedback extension, in which estimates of SNR were used to adjust feature detector sensitivities, a different pattern of improvements were seen. For example, "wheek"-detection in noise was now improved (unlike with contrast gain control).

Finally the authors adapt the model to implement it with "biologically-plausible" mechanisms inspired by theories of neural circuits. They compare the performance of this model with behavioral experiments with Guinea pigs, and make the claim that only models with both adaptation types (contrast gain, and attention mediated feedback) can account for observed Guinea pig behavior.

Overall, the authors have picked a very interesting and relevant research question, and I believe their approach of adapting and extending models to be noise-robust and comparing with measured behavior is a very powerful tool to explore such questions. I believe this work deserves eventual publication and will be a worthy contribution to the field.

* == Major Issues ==

Unfortunately, I do have several critiques of the manuscript in its current form and I cannot endorse its publication until these are addressed:

1. I find the comparison between model and behavior unconvincing. The authors show results for two different calls and for one (the "wheek") all models (even those without adaptation) outperform the guinea pig. For the other call (the "chut") the previously present model cannot account for behavior unless it is augmented with contrast gain control (for low-noise levels) or attentional feedback (for high-noise levels).

Thus, the resounding conclusion is that this model recapitulates observed biological hearing sometimes (i.e., for "chuts") but not always (for "wheeks" none of this seems important). Of course, all models have limits, but in order to learn from them we must ask why this is so? And here the authors do not do enough to answer this question, in my opinion.

I see two possible solutions. The authors could obtain behavioral data for many more vocalizations (the marmoset "twitter" and "trill"?) and with reverberation, and demonstrate that the adaptive models are required to recapitulate observed behavior for *most cases*. (as opposed to half the cases at best, in the current data).

Or, alternatively, the authors could give a reader some explanation for why we might expect adaptation to be more important for "chuts" as opposed to "wheeks". The authors hint at this but don't give many details. This leads me directly to my next critique.

2. The authors do not tell us enough about the stimuli for readers to interpret the work. In plots 2, 4, and 6, we see that categorization models perform differently on 4 different vocalization patterns and with different types of noise and reverberation. Why is this so? Presumably the structure of the sources and reverberation are important but the authors don't tell us what these structures are. Perhaps they are published in prior papers but I believe this information is too important to send a reader searching. I think the authors need to show cochleagrams of examples of all the vocalization types. And at least some of them with the applied reverberation.

It is to be expected that reverberation will obscure some signals (short-transient signals with silent-pauses) more than others (continuous pitched signals). Likewise, very sparse signals, are likely more robust to noise than dense signals (with the same SNR). Which type of signals are being investigated here? Perhaps the behavioral result that "hearing chuts" requires neural adaptation but "hearing wheeks" does not has a perfectly reasonable explanation. I believe readers need to see the structure of the calls (and the reverberation) to interpret this work.

I would suggest adding cochleagrams of each vocalization type to figure 1 or 2. And I would suggest adding example cochleagrams of the "wheel" and "chut" (which are used in the behavioral experiment) with the different reverberation conditions in Figure 2, or at the very least in the supplemental information.

3. As the authors say very little of substance about acoustic reverberation, I would suggest they remove the word from the title. Or if they wish to keep it prominent they should write more about the structure of the reverberation they used.

In figures 2, 4, and 6, we see model categorization performance in the presence of 6 different types of reverberation, we must refer to the methods to learn where this reverberation came from and all we learn are some category labels and T30 measures. In figure 3 we see a cochleagram with a "twitter" with and without noise, but we do not see it with reverberation. If the 8 categories of reverberation were replaced with 8 different manipulations (e.g. interference from other sources, truncations, clipping, or band-pass filtering) the content of the paper would barely change. There would be some changes to model performance, the adaptive model would probably perform better in some cases, but not in others, and the reader would still have no idea why.

Moreover, the contrast gain control mechanism is justified as a mechanism to provide robustness to noise (as the authors describe: noise increases mean and decreases variance). Does reverberation also increase the cochleagram mean and decrease variance? Figure 3 could show this, but it does not. So, the mechanism is justified for noise, and just happens to (sometimes) work in reverberation.

Similarly, the top-down adaptation shows a simple structure with noise (optimal thresholds scale linearly with SNR) but not with reverberation. The authors do not say why.

Finally, and most significantly, in my opinion, reverberation is not considered in the behavioral experiments. All in all, this paper writes intelligently about noise but treats reverberation as an afterthought. This is a perfectly reasonable thing to do, as hearing in noise is a significant problem of its own. But I would suggest the paper title be amended to reflect this focus on noise.

* == Recommendation ==

If the authors address the 3 major issues listed above, I would consider approving this manuscript for publication. The work is interesting and, with more information provided, I believe the field will benefit from these results.

* == Minor Issues ==

The remainder of this review will list minor issues:

- as I read the section on contrast gain control I wasn't sure if the "top-down" adaptive thresholds (mentioned already in Figure 1) were considered a part of this model or not. Readers may benefit from a statement clarifying that the adaptive threshold will be considered in the next model.

- as I read the section on "top-down adaptation" I noted that the auditory system does not have the benefit of knowing SNR in advance. This is addressed in the next section but readers would benefit from this being acknowledged early on, so we know that a fix is coming. That would have saved me re-reading the section several times to see if I had missed something.

Reviewer #3 (Remarks to the Author):

The study by Parida et al. analyzed an extension of the theoretical model, which was able reliably and invariantly categorized animal vocalizations based on a relatively small set of feature characteristics extracted from spectrotemporal signal representations. Authors extended the original model using biologically relevant implementation of unsupervised modulation of the model function and demonstrated that this model can substantially improve performance of the model in complex listening conditions. The results of this complex model mainly corresponds to what we would expect based on the animal performance in similar condition. The model thus proved to be relevant for testing different mechanisms of bottom-up and to-down control to achieve robust feature extraction and categorization in the whole auditory system.

The present study demonstrated that their model provides potentially important and powerful, yet comparatively simple, tool for expanding our understanding of the function and adaptation in auditory system. The results are presented conclusively and the manuscript is written clearly. Before I can fully recommend it to publication, I have couple of points that should be addressed.

MAJOR COMMENTS:

- The biological interpretations of the implemented gain control mechanisms and especially top-down control, as they are presented in the manuscript, should be expanded or weakened, so that it would be explicitly clear that they serve as examples of these mechanisms. Although attention and arousal are behavioral features that involve activation of top-down processes, they are, by far, not the only ones. E.g. cholinergic, noradrenergic, dopaminergic or serotonergic modulation are extremely important signals for enhancing plastic adaptation of neuronal neural networks to behaviorally relevant changes in the environment. This diversity seems to be very important for robustness of biological neuronal networks. They can change excitability of neurons in short- and/or long-term manner. Many of these processes can be realistically implemented by threshold shifts (as it was used in this study)

and/or change the slope of activation function. Consequently, these neuronal mechanisms could be relevantly studied by the present model, which would increase its importance. To put it clear: I am not expecting to describe all known mechanisms for top-down control, I would only like to encourage author to explicitly express that they are putting examples of the biological mechanism.

- How the present model capture temporal dynamics of the sound signal? The dynamics of temporal sequences plays substantial role in sound and speech categorization. Classical examples of categorical speech processing, voiced-onset time, is based on nonlinear neuronal processing of temporal sequences in both human and animals. Subcortical processing of temporal sequences seems to be important for sound categorization, even in animals (also in guinea pigs). I am wondering if such seemingly important mechanisms can be incorporated in your model, or you felt that they can emerge from your current implementation. Short discussion of this aspect could be beneficial especially in the context of categorization.
- In the discussion part dealing with task-specific training results, authors come to the conclusion that '...task-specific training is an inefficient way to categorize sounds.' Results are, however, indicating the opposite if I didn't miss something important. Except of a few exceptions, the model trained in noise or reverberations environment outperforms models trained using clean calls only. Obviously, it is to be expected that the model would optimize its performance if trained in mixed environment (as authors has indicated in Result section). It's a pity that authors did not include such training condition in the current study and did not present mixed-trained model performance. It could provide us with an evidence and would not let us with a hypothesis. Anyway, I would encourage author to expand this section in Discussion, perhaps including similar way of argumentation and inference as they have shortly provided in the Result section.
- In real world, the recognition and categorization of the sounds is influenced by both gain and top-down control, i.e. both adaptations act simultaneously. It would be optimal if we can see also the results of this adapted model. This would also provide more insight how these mechanisms interact in the model. It would be especially interesting to see comparison of results of this model with behavioral results. Alternatively, authors can, at least, discuss such interactions.

MINOR COMMENTS:

- Authors can consider to introduce abbreviation for SNR, even being generally known, to exclude any confusions.
- I would also encourage authors to present example cochleagrams of all analyzed vocalizations (Twitter, Trill, Chut and Wheel), perhaps in Supplementary Information. It would make easier for reader to have a direct access to this information within single paper, and try to understand differences in the classifier performances in different conditions.
- I would also appreciate more detailed description of the basics of the procedure used for generation of reverberant calls.
- Page 6: 'To ensure that our results reflected general auditory processing principles, we trained and tested our models on calls from two different species...' In my understanding, you cannot ensure generality of the principle from two examples. You can, perhaps, imply it. Analyzing calls from two species is definitely better than from the single one, and I appreciate that. Ensuring generality requires, however, more than two.
- processing principles, we trained and tested our models on calls from two different species
- Fig. 3A: description of the axes for cochleagrams should be consistent.
- Missing y-axis legend in Fig. 7A top. It is most probably the same as the axis below, but to repeat it or shift it so that it would clearly belong to both y-axis would help.
- In Fig. 7B, C and CGC is not defined (CGC is introduced in Fig. 3, it could be for better understanding repeated here as well, or introduced in text).
- Fig. 7C, color coding is referring to the color of the boxes in B, but explicit legend would help.
- Fig. 5: 'The distribution of feature-detector responses linearly scaled with SNR...'. SNR is a logarithmic scale and y-axis is linear. Consequently, linear relationship in semi logarithmic scale would imply exponential relation and not linear. The main outcome has, however, not changed – Optimal ration depends on SNR, but not on reverb type.

- Supplementary Tables (Table S1 and S2) would, in my view, benefit from adding more elaborative captions.

We thank the two reviewers for carefully evaluating this manuscript and for their constructive comments, which we feel have led to an improved manuscript after this revision. The revised manuscript includes new behavioral experiments, new modeling, and additional analyses in response to reviewer comments, presented in four new figures (two main, two supplemental). In the response to reviewers below, original reviewer comments are in blue, and our responses are in black. Page/Line/Figure numbers in blue refer to the original submission, whereas page/line/figure numbers in black refer to the clean revised version.

Reviewer #1 (Remarks to the Author):

* == Summary ==

The authors consider a model of robust auditory categorization. In particular, they extend a previous model to provide robustness to variability from the external environment, such as due to noise and signal reverberation. The authors consider three different model extensions, and compare each with task behavior observed in guinea pigs. The authors claim that models with adaptive mechanisms show better categorization, and produced responses most similar to the guinea pig responses, and thereby conclude that such adaptive mechanisms are important for robust auditory categorization.

Each of the three model extensions resulted in improved categorization, but the improvements were greater for some vocalization types, and some noise types than others, and this pattern of improvement varied substantially across the three different extensions.

Specifically:

A. For the condition-specific training, training in noise aided recognition in noise, training in reverberation aided recognition in reverberation. There was little evidence of a generalized benefit from either. Also, the authors note this is an expensive solution. New features must be learned and remembered for each noisy environment.

B. For the contrast gain control, in which the signal was de-measured and variance-normalized before feature detection, performance improved for both recognition of noisy and reverberant calls. However, very different improvements were observed for different call types. For example, "twitter"-detection was much improved but "wheel"-detection was not.

C. For the "top-down" mediated feedback extension, in which estimates of SNR were used to adjust feature detector sensitivities, a different pattern of improvements were seen. For example, "wheel"-detection in noise was now improved (unlike with contrast gain control).

Finally the authors adapt the model to implement it with "biologically-plausible" mechanisms inspired by theories of neural circuits. They compare the performance of this model with behavioral experiments with Guinea pigs, and make the claim that only models with both adaptation types (contrast gain, and attention mediated feedback) can account for observed Guinea pig behavior.

Overall, the authors have picked a very interesting and relevant research question, and I believe their approach of adapting and extending models to be noise-robust and comparing with measured behavior is a very powerful tool to explore such questions. I believe this work deserves eventual publication and will be a worthy contribution to the field.

We thank the reviewer for their enthusiasm for our work and their constructive feedback.

* == Major Issues ==

Unfortunately, I do have several critiques of the manuscript in its current form and I cannot endorse its publication until these are addressed:

1. I find the comparison between model and behavior unconvincing. The authors show results for two different calls and for one (the "wheek") all models (even those without adaptation) outperform the guinea pig. For the other call (the "chut") the previously present model cannot account for behavior unless it is augmented with contrast gain control (for low-noise levels) or attentional feedback (for high-noise levels).

Thus, the resounding conclusion is that this model recapitulates observed biological hearing sometimes (i.e., for "chuts") but not always (for "wheeks" none of this seems important). Of course, all models have limits, but in order to learn from them we must ask why this is so? And here the authors do not do enough to answer this question, in my opinion.

I see two possible solutions. The authors could obtain behavioral data for many more vocalizations (the marmoset "twitter" and "trill"?) and with reverberation, and demonstrate that the adaptive models are required to recapitulate observed behavior for **most cases**. (as opposed to half the cases at best, in the current data).

Or, alternatively, the authors could give a reader some explanation for why we might expect adaptation to be more important for "chuts" as opposed to "wheeks". The authors hint at this but don't give many details. This leads me directly to my next critique.

We agree with the reviewer that the comparison to behavior in the original manuscript was based on limited behavioral data. We have tried to address this concern in three ways. First, in the few months following our receipt of these comments, we have obtained behavioral data for reverberant conditions (for GP calls) to add to the model-data comparisons (expanded Figure 10). We could not obtain marmoset behavioral data as our lab no longer directly works with this species (vocalizations were obtained from the lab of Xiaoqin Wang for the modeling studies, see *Acknowledgements*).

Second, regarding the differential effects across call types, we realized that directly comparing model outputs to behavioral results might be misleading as this does not account for stimulus-independent effects (motivation, arousal, etc.) that may be different between different subjects used in the experiments with different call types. Therefore, based partly on a previous study from our lab (Kar et al., 2022), we implemented a method of adjusting the model output so that it can be directly compared to behavioral data. Briefly, we learned an additional mapping to equalize model and behavioral performance in the clean condition and used this mapping to adjust model outputs in the noisy and reverberant conditions (Fig. S4). This approach has been used in several speech perception studies (Relaño-Iborra et al., 2016; Viswanathan et al., 2021). With this adjustment, the effects of the different adaptive mechanisms are more consistent across call types (Fig. 10).

Finally, to better understand why different adaptive mechanisms work better in different noisy and reverberant conditions, we performed new analyses where we explored the benefit of including adaptive mechanisms on model performance as a function of reverberation strength or SNR level (Fig. 8). To our surprise, the benefit due to adaptive mechanisms was not correlated with reverb strength or

SNR; rather, they were correlated with how well the original model was performing in the original condition. If the original model's performance was high, adding adaptive mechanisms had little additional benefit (ceiling effect). If the original model's performance was abysmal, the adaptive mechanisms did not help at all. But for conditions where the original model's performance was moderate, adaptive mechanisms could confer a sizeable benefit. This is discussed in new Figure 8 and associated main text (L375-409).

Taken together, we believe that the additional behavioral data, adjustments to the model, and additional figure now provide a much deeper explanation for the effects of the adaptive mechanisms explored in this study.

2. The authors do not tell us enough about the stimuli for readers to interpret the work. In plots 2, 4, and 6, we see that categorization models perform differently on 4 different vocalization patterns and with different types of noise and reverberation. Why is this so? Presumably the structure of the sources and reverberation are important but the authors don't tell us what these structures are. Perhaps they are published in prior papers but I believe this information is too important to send a reader searching. I think the authors need to show cochleagrams of examples of all the vocalization types. And at least some of them with the applied reverberation.

It is to be expected that reverberation will obscure some signals (short-transient signals with silent-pauses) more than others (continuous pitched signals). Likewise, very sparse signals, are likely more robust to noise than dense signals (with the same SNR). Which type of signals are being investigated here? Perhaps the behavioral result that "hearing chuts" requires neural adaptation but "hearing wheeks" does not has a perfectly reasonable explanation. I believe readers need to see the structure of the calls (and the reverberation) to interpret this work.

I would suggest adding cochleagrams of each vocalization type to figure 1 or 2. And I would suggest adding example cochleagrams of the "wheel" and "chut" (which are used in the behavioral experiment) with the different reverberation conditions in Figure 2, or at the very least in the supplemental information.

Thank you for this suggestion. We have added a figure (new Fig. 2), which shows one example of each call type used in this study in clean condition as well as in a noisy and a reverberant condition. We have also described in more detail the spectrotemporal properties of these call types in Fig. 2 caption as well as in the text (L149-163), touching on many of the points raised by the reviewer in their comments. Finally, in new Fig. 8 and associated text (L375-409), we explore possible explanations for the differential effects of the mechanisms.

3. As the authors say very little of substance about acoustic reverberation, I would suggest they remove the word from the title. Or if they wish to keep it prominent they should write more about the structure of the reverberation they used.

In figures 2, 4, and 6, we see model categorization performance in the presence of 6 different types of reverberation, we must refer to the methods to learn where this reverberation came from and all we learn are some category labels and T30 measures. In figure 3 we see a cohleagram with a "twitter" with and without noise, but we do not see it with reverberation. If the 8 categories of reverberation were replaced with 8 different manipulations (e.g. interference from other sources, truncations, clipping, or band-pass filtering) the content of the paper would barely change. There would be some changes to

model performance, the adaptive model would probably perform better in some cases, but not in others, and the reader would still have no idea why.

Moreover, the contrast gain control mechanism is justified as a mechanism to provide robustness to noise (as the authors describe: noise increases mean and decreases variance). Does reverberation also increase the cochleagram mean and decrease variance? Figure 3 could show this, but it does not. So, the mechanism is justified for noise, and just happens to (sometimes) work in reverberation.

Similarly, the top-down adaptation shows a simple structure with noise (optimal thresholds scale linearly with SNR) but not with reverberation. The authors do not say why.

Finally, and most significantly, in my opinion, reverberation is not considered in the behavioral experiments. All in all, this paper writes intelligently about noise but treats reverberation as an afterthought. This is a perfectly reasonable thing to do, as hearing in noise is a significant problem of its own. But I would suggest the paper title be amended to reflect this focus on noise.

Thank you for these important suggestions. We believe that the reverberation aspect of our study is important to warrant inclusion in the title, but agree that the original manuscript did not provide due importance to reverberation. We have remedied this in the revised manuscript in three ways:

1. We have added a supplemental figure (new Fig. S1) that describes the reverberation impulse responses (time and frequency domain) as well as the power decay function that we used to estimate T30 values. We also highlight that the effect of noise is additive whereas that of reverberation is convolutional. Therefore, noise and reverberation can be more detrimental depending on the spectrotemporal structure of the stimulus (L149-163).
2. We have now added the effect of reverberation to (new) Fig. 4 and state that reverberation also increases the mean and reduces the standard deviation at the FD input stage (L251-255).
- 3, We have acquired behavioral data for four (of eight) reverberant conditions and the data are shown in (new) Fig. 9.

Unfortunately, despite much effort, we presently do not have an explanation for why the optimal threshold ratio does not scale with reverberation. A possible but unlikely explanation could be that the range of reverberation T30s we have used in this study (7 ms to 644 ms) is too small to show such scaling. We will follow up on this in future studies, but for now, we have included a statement in the discussion that we cannot explain this result (L569-572).

* == Recommendation ==

If the authors address the 3 major issues listed above, I would consider approving this manuscript for publication. The work is interesting and, with more information provided, I believe the field will benefit from these results.

* == Minor Issues ==

The remainder of this review will list minor issues:

- as I read the section on contrast gain control I wasn't sure if the "top-down" adaptive thresholds (mentioned already in Figure 1) were considered a part of this model or not. Readers may benefit from a statement clarifying that the adaptive threshold will be considered in the next model.

We now mention that the two models considered in the contrast gain control section do not have top-down modulation (L296).

- as I read the section on "top-down adaptation" I noted that the auditory system does not have the benefit of knowing SNR in advance. This is addressed in the next section but readers would benefit from this being acknowledged early on, so we know that a fix is coming. That would have saved me re-reading the section several times to see if I had missed something.

Thank you for the comment. We now acknowledge this issue in the top-down modulation section (~L326-330), and mention that a biologically feasible implementation is considered later in the manuscript.

Reviewer #3 (Remarks to the Author):

The study by Parida et al. analyzed an extension of the theoretical model, which was able to reliably and invariantly categorize animal vocalizations based on a relatively small set of feature characteristics extracted from spectrotemporal signal representations. Authors extended the original model using biologically relevant implementation of unsupervised modulation of the model function and demonstrated that this model can substantially improve performance of the model in complex listening conditions. The results of this complex model mainly correspond to what we would expect based on the animal performance in similar condition. The model thus proved to be relevant for testing different mechanisms of bottom-up and top-down control to achieve robust feature extraction and categorization in the whole auditory system. The present study demonstrated that their model provides a potentially important and powerful, yet comparatively simple, tool for expanding our understanding of the function and adaptation in the auditory system. The results are presented conclusively and the manuscript is written clearly. Before I can fully recommend it to publication, I have a couple of points that should be addressed.

We thank the reviewer for their evaluation and the thorough and constructive feedback.

MAJOR COMMENTS:

- The biological interpretations of the implemented gain control mechanisms and especially top-down control, as they are presented in the manuscript, should be expanded or weakened, so that it would be explicitly clear that they serve as examples of these mechanisms. Although attention and arousal are behavioral features that involve activation of top-down processes, they are, by far, not the only ones. e.g. cholinergic, noradrenergic, dopaminergic or serotonergic modulation are extremely important signals for enhancing plastic adaptation of neuronal neural networks to behaviorally relevant changes in the environment. This diversity seems to be very important for robustness of biological neuronal networks. They can change excitability of neurons in short- and/or long-term manner. Many of these processes can be realistically implemented by threshold shifts (as it was used in this study) and/or change the slope of activation function. Consequently, these neuronal mechanisms could be relevantly studied by the present model, which would increase its importance. To put it clear: I am not expecting to

describe all known mechanisms for top-down control, I would only like to encourage author to explicitly express that they are putting examples of the biological mechanism.

We agree – what we highlighted in terms of VIP-mediated disinhibition is one of many possible mechanisms by which top-down control can be exerted on auditory cortical responses. In retrospect, perhaps this was too strongly stated in the original manuscript. In the revision, we are careful to mention that this is one example, and also mention alternative mechanisms. Similarly, we mention alternative mechanisms of implementing contrast gain control, and state the experimental data that support (or do not support) these mechanisms. For example, see lines L547-556, L576-588.

- How does the present model capture temporal dynamics of the sound signal? The dynamics of temporal sequences plays a substantial role in sound and speech categorization. Classical examples of categorical speech processing, voice-onset time, is based on nonlinear neuronal processing of temporal sequences in both humans and animals. Subcortical processing of temporal sequences seems to be important for sound categorization, even in animals (also in guinea pigs). I am wondering if such seemingly important mechanisms can be incorporated in your model, or you feel that they can emerge from your current implementation. Short discussion of this aspect could be beneficial especially in the context of categorization.

Thank you for this question. The model already captures temporal sequence information of the order of ~100 ms because the feature detectors (FDs) typically have 1 – 2 octave bandwidths and 100 – 200ms durations. Thus, the FD STRFs typically include multiple frequency components and syllables of the calls. When we convolve the FD STRFs with the stimulus, we are essentially looking for temporal sequences in this range of bandwidth and time. We have previously shown that the receptive fields of A1 L2/3 neurons are consistent with such FD STRFs and show structures that include several components of the calls (Montes-Lourido et al., 2021).

Several known neuronal tuning profiles can emerge from such FD STRFs. For example, we have previously shown that the model can capture aspects of neural tuning properties that are not explained by simple spectral tuning (e.g., asymmetric sensitivity to upward and downward frequency modulation; Fig. 8 in Liu et al., 2019). Similarly, band-pass modulation tuning, which is typical of midbrain and cortical neurons, can also emerge from the model. For example, in Fig. R1 we show the maximum Vm response of a rumble FD in response to amplitude-modulated white Gaussian noise with different AM frequencies. The FD shows a bandpass modulation transfer function sharply tuned to 16 Hz AM, which arises from the fact that the FD was selective for multiple syllables of the rumble call, which has this modulation present. Therefore, the FDs already capture some specific spectrotemporal features that are informative in categorizing a stimulus type.

Fig. R1: Modulation tuning can emerge in model feature detectors. **Left** The spectrotemporal receptive field of an example rumble FD. **Right** Maximum Vm response to amplitude-modulated white Gaussian noise at different modulation frequencies.

Our model does not capture longer time-scale sequences i.e., as currently implemented, the temporal order of the detected features does not matter for final categorization. Rather, we weight and integrate the detected features without considering the time or order of detection and perfectly integrate these without temporal decay. As we have shown in previous publications, for animal calls, this appears to sufficiently account for behavioral responses. But future implementations of the model will consider these longer-term temporal dependencies to categorize more complex stimuli, such as speech for example. We have included a brief statement in the introduction section explaining these issues when introducing the model (L64-66 and L69-71).

- In the discussion part dealing with task-specific training results, authors come to the conclusion that ‘..,task-specific training is an inefficient way to categorize sounds.’ Results are, however, indicating the opposite if I didn’t miss something important. Except of a few exceptions, the model trained in noise or reverberations environment outperforms models trained using clean calls only. Obviously, it is to be expected that the model would optimize its performance if trained in mixed environment (as authors has indicated in Result section). It’s a pity that authors did not include such training condition in the current study and did not present mixed-trained model performance. It could provide us with an evidence and would not let us with a hypothesis. Anyway, I would encourage author to expand this section in Discussion, perhaps including similar way of argumentation and inderence as they have shortly provided in the Result section.

Thank you for the recommendation - this is indeed an important condition to consider. In the revised manuscript, we now include performances of models trained on a mix of noisy and reverberant calls (new Fig. 3). Model training in mixed conditions indeed outperformed models trained on a single condition (including when training and testing conditions were the same). This is likely due to the model optimizing its features (as you mentioned), which could help to reduce model overfitting. The implications of this result are also discussed (L515-532).

- In real world, the recognition and categorization of the sounds is influenced by both gain and top-down control, i.e. both adaptations act simultaneously. It would be optimal if we can see also the results of this adapted model. This would also provide more insight how these mechanisms interact in the model. It would be especially interesting to see comparison of results of this model with behavioral results. Alternatively, authors can, at least, discuss such interactions.

Thank you for the suggestion. We have included results for a model that includes both CGC and top-down (new Fig. 7). While including both mechanisms generally improves model performance compared to when only one mechanism is included (χ^2 values in Table 2), the improvement is sublinear (new Fig. 8). In other words, improvement due to including both mechanisms (compared to the original model without any mechanisms) is less than the sum of improvement due to either mechanism alone. We can in fact demonstrate that the effects of these adaptations differentially improve performance in different SNR ranges.

MINOR COMMENTS:

- Authors can consider to introduce abbreviation for SNR, even being generally known, to exclude any confusions.

We have introduced SNR in Fig 2 as well as in the first occurrence in text (L157).

- I would also encourage authors to present example cochleagrams of all analyzed vocalizations (Twitter, Trill, Chut and Wheek), perhaps in Supplementary Information. It would make easier for reader to have a direct access to this information within single paper, and try to understand differences in the classifier performances in different conditions.

We have added a figure (new Fig. 2), which includes exemplar cochleagrams of all call types used in this study. We have also described in more detail the spectrotemporal properties of these call types in Fig. 2 caption as well as in the text (L149-163).

- I would also appreciate more detailed description of the basics of the procedure used for generation of reverberant calls.

Thank you for these important suggestions. We have added a supplemental figure (new Fig. S1) that describes the reverberation impulse responses (time and frequency domain) as well as the power decay function that we used to estimate T30 values. We also highlight that the effect of noise is additive whereas that of reverberation is convolutional. Therefore, noise and reverberation can be more detrimental depending on the spectrotemporal structure of the stimulus (L159-163).

- Page 6: 'To ensure that our results reflected general auditory processing principles, we trained and tested our models on calls from two different species...' In my understanding, you cannot ensure generality of the principle from two examples. You can, perhaps, imply it. Analyzing calls from two species is definitely better than from the single one, and I appreciate that. Ensuring generality requires, however, more than two. • processing principles, we trained and tested our models on calls from two different species

Fair point – we have removed that wording.

- Fig. 3A: description of the axes for cochleagrams should be consistent.

We have updated this figure (new Fig. 4).

- Missing y-axis legend in Fig. 7A top. It is most probably the same as the axis below, but to repeat it or shift it so that it would clearly belong to both y-axis would help.

Updated (new Fig. 9A).

- In Fig. 7B, C and CGC is not defined (CGC is introduced in Fig. 3, it could be for better understanding repeated here as well, or introduced in text).

We have reintroduced CGC here, and generally introduced any abbreviations used in a figure in its caption.

- Fig. 7C, color coding is referring to the color of the boxes in B, but explicit legend would help.

Updated (new Fig. 8C).

- Fig. 5: ‘The distribution of feature-detector responses linearly scaled with SNR...’. SNR is a logarithmic scale and y-axis is linear. Consequently, linear relationship in semi logarithmic scale would imply exponential relation and not linear. The main outcome has, however, not changed – Optimal ratio depends on SNR, but not on reverb type.

Thank you for the technical comment – we agree. We have added “(in dB)” to clarify that linear relation between optimal ratio applies to SNR in dB scale in text (L330) and new Fig. 8 caption.

- Supplementary Tables (Table S1 and S2) would, in my view, benefit from adding more elaborative captions.

We have added captions to these supplemental tables.

References

- Kar, M., Pernia, M., Williams, K., Parida, S., Schneider, N.A., McAndrew, M., Kumbam, I., Sadagopan, S., 2022. Vocalization categorization behavior explained by a feature-based auditory categorization model. *eLife* 11, e78278. <https://doi.org/10.7554/eLife.78278>
- Liu, S.T., Montes-Lourido, P., Wang, X., Sadagopan, S., 2019. Optimal features for auditory categorization. *Nat Commun* 10, 1302. <https://doi.org/10.1038/s41467-019-09115-y>
- Montes-Lourido, P., Kar, M., David, S.V., Sadagopan, S., 2021. Neuronal selectivity to complex vocalization features emerges in the superficial layers of primary auditory cortex. *PLOS Biology* 19, e3001299. <https://doi.org/10.1371/journal.pbio.3001299>
- Relaño-Iborra, H., May, T., Zaar, J., Scheidiger, C., Dau, T., 2016. Predicting speech intelligibility based on a correlation metric in the envelope power spectrum domain. *The Journal of the Acoustical Society of America* 140, 2670–2679. <https://doi.org/10.1121/1.4964505>
- Viswanathan, V., Bharadwaj, H.M., Shinn-Cunningham, B.G., Heinz, M.G., 2021. Modulation masking and fine structure shape neural envelope coding to predict speech intelligibility across diverse listening conditions. *The Journal of the Acoustical Society of America* 150, 2230–2244. <https://doi.org/10.1121/10.0006385>

REVIEWERS' COMMENTS:

Reviewer #1 (Remarks to the Author):

I am satisfied that the authors have addressed all the major issues I raised on the first submission. The information provided in the new figures makes the paper much more interesting for a reader. I endorse this paper for publication.

My only remaining comments are minor issues:

- + All cochleograms (Figs 1, 2, 3)
- + All cochleograms should have a colorbar.
- + I would recommend a different colormap. For spectrotemporal power a "sequential" colormap (e.g. <https://matplotlib.org/stable/tutorials/colors/colormaps.html#sequential>) results in colors that faithfully represent the variable plotted to the human eye. And those that fade to white at the low-end are convenient for an "inaudible" threshold.

- + Fig 2.
 - + in the marmoset calls there seems to be low-frequency noise in the "clean" recordings. Is this background noise from when the recordings were made? If so, a reader may like to know that the model performance doesn't change if this is filtered out prior to the analyses you have done. (a sentence or two in the results section will do). If it is actually a complex part of the vocalization the figure caption should reference this so a reader doesn't think it's noise.

- + Fig 8.
 - + some readers may appreciate the legend showing which colors correspond to which call types.

- + Fig S1
 - + in (b) some envelopes show wraparound artifacts with reverberant power increasing at the very end. You should zero-pad the signal before applying a Hilbert-transform (or any Fourier domain processing)

Reviewer #3 (Remarks to the Author):

The authors have adequately addressed all major points I raised in previous comments. After these revisions, I can recommend the manuscript for publication in Communications Biology.

Nevertheless, I would like to make only two minor comments:

- I think that the combination of both mechanisms shows the increased robustness of the model, even it did not supralinearly boost model performance. The two mechanisms implemented in the current model did not interact destructively, which is important.
- In Fig. 2, authors may consider increasing the dynamic range of the cochleograms. This would make it easier to recognize the structure of the vocalizations.